# Standard ingredient of *Drosophila* medium reduces transmission and virulence of the gut pathogen *Pseudomonas entomophila*

Youn Henry,[1] Berta Canal-Domènech,[1] Jaime González,[1] Christine La Mendola,[1] Tadeusz J. Kawecki[1]

**ABSTRACT** In the last 20 years, *Pseudomonas entomophila* (Pe) has emerged as a model to explore insect immunity to bacterial intestinal pathogens. Laboratory studies have characterized multiple detrimental effects of Pe on *Drosophila melanogaster*. However, these effects require that the bacteria are ingested in extremely high concentrations of $10^{10}$–$10^{11}$ CFU per mL ($OD_{600}$ 20–200), questioning the relevance of this pathogen in nature. Here, we tested whether the need for such high doses may be due to protective effects of the antifungal agent methylparaben (Nipagin), a standard ingredient of laboratory *Drosophila* diets. While significant mortality of flies fed diet containing methylparaben required pathogen concentrations of $>10^{10}$ CFU per mL, we could induce mortality with 500,000-fold lower dose when methylparaben was absent. Here, even this small infection dose ($10^5$ CFU per mL) led to high bacterial loads ($10^6$ CFU per fly) after several days, indicating the ability of Pe to grow and overcome the flies' defenses in the absence of methylparaben. Consistent with these results, we show strong bactericidal properties of methylparaben on Pe *in vitro*. We also demonstrate that, in the absence of methylparaben, infected flies can easily transmit the pathogen to other adults and to offspring, resulting in high mortality and thus highlighting the potential of Pe as a pathogen of *Drosophila* in nature. For those reasons, careful consideration should be given to food additives used in standard diets in laboratory research on host-pathogen interactions.

**IMPORTANCE** Accurate characterization of pathogen infections requires appropriate experimental methodologies. Infections of insects with Pe are frequently studied using fruit flies as a model organism, with laboratory cultures typically maintained on artificial media containing various food preservatives. In this study, we show that one commonly used preservative, methylparaben, significantly influences the outcome of oral infections with Pe. We found that minimal infection doses, far below the standards of the field, could still be lethal to flies raised on media without methylparaben. This increased virulence was also associated with increased transmission of the pathogen, both from infected adult flies to their offspring and to uninfected adults. Our findings show how subtle variations in experimental conditions can profoundly affect how we perceive pathogenic threats.

**KEYWORDS** *Drosophila melanogaster*, insects, bacteria, host-pathogen interactions, methylparaben, Nipagin, antifungal

Address correspondence to Youn Henry, youn.henry@unil.ch.

The authors declare no conflict of interest.

See the funding table on p. 13.

Since Vodovar et al. (1) first described *Pseudomonas entomophila* (Pe) and characterized its pathogenicity for fruit flies, this Gram-negative Gammaproteobacterium has been widely used as a model in fields such as immunity, ecology, evolution, or sexual selection (2–6). Although Pe can infect multiple insect species, much research has focused on its interaction with the fruit fly *Drosophila melanogaster* (3). When ingested in sufficiently high doses, the toxins released by this pathogen in conjunction with reactive

oxygen species (ROS) produced by the fly itself as part of its immune response lead to the rupture of the gut epithelium and eventually to death of the flies (1, 2, 7, 8).

One peculiar feature of most studies that performed oral infections of Pe is the dose of pathogen used (see Table S1 for an overview of doses used in different studies). Following the example of the first publication on the topic (1), researchers usually feed the flies highly concentrated Pe suspensions (optical density at 600 nm 20–200), resulting in ingested loads of $10^5$–$10^8$ CFU per fly within a day (7, 9, 10). This practice does not reflect any ecologically realistic situation, as such concentrations can only be obtained artificially, by centrifugation. The need to use high oral doses is even more surprising when compared to the low doses used for systemic infections by pricking with a needle, which are lethal even with inocula of *c.a.* 50 bacteria per fly (8). The protective function of the epithelial gut barrier in conjunction with the peritrophic matrix—a semi-permeable layer secreted in the midgut to encapsulate food—is the main argument used to explain the high doses required to cause mortality upon oral infection (7, 11–13). As a result, pathogen clearance in the gut of fruit flies has been reported to occur in less than 16 h and has been attributed to an efficient immune system (1, 7, 14). However, this explanation might only be part of the story.

All published experimental studies working with the Pe-*Drosophila* system have been performed in the laboratory, with flies and larvae fed artificial diet media composed of agar, nutrients (yeast, sugar, often cornmeal, or another source of starch), and antifungal preservatives. These preservatives, such as propionic acid but primarily methylparaben (CAS No. 99-76-3, hereafter named "mp" and alternatively named Nipagin, Tegosept, or Moldex in the literature), attracted our attention. Although researchers use them to avoid mold in batch fly cultures, potential side effects for flies or for fly-associated bacteria have been largely unexplored. While antibacterial properties of mp have been described decades ago (e.g., 15–18), it is only recently that several studies explicitly pointed out the important effects of mp on growth of fly microbiota (19–21). In particular, mp caused marked growth inhibition of *Acetobacter* and, to a lesser extent, of *Lactiplantibacillus*, two abundant commensal microbiota species of fruit flies. In fact, mp is likely responsible for conflicting results reported in studies investigating the effect of microbiota on sexual behavior (22–25). Finally, we noted that the only two studies that employed low Pe infection doses were also the ones not using any antifungal preservative (11, 26) (Table S1). All these elements point to the hypothesis that mp also exerts negative effects on Pe, thereby explaining the unrealistically high pathogen doses required to harm flies.

In this study, we first tested whether mp (as well as propionic acid, the second preservative often used in *Drosophila* media) affected Pe growth *in vitro*. Then, to assess the protective effect of mp on flies exposed to Pe, we subjected flies maintained on a diet with and without mp to oral infection at various doses and quantified their survival and Pe load. The results demonstrated that in the absence of antimicrobial agents, Pe is highly virulent at doses six orders of magnitude lower than those typically used in past studies. Therefore, we also investigated the potential for direct and indirect adult-to-offspring and adult-to-adult transmission, exploring the consequences of our findings on the ecology of this insect-pathogen relationship.

## RESULTS

### Methylparaben and propionic acid are bactericidal to Pe *in vitro*

We tested how different mp doses harm Pe growth (minimum inhibitory concentration, MIC) or viability (minimum bactericidal concentration, MBC) *in vitro*. We found that 0.2% was both the MIC and MBC of mp on Pe, as no growth could be observed during 22 h of incubation, and no viable colonies were recovered when the final suspension was plated. Concentrations of 0.002% and lower all showed normal growth (area under curves = 12.6 [12.3; 12.8]), and 0.02% suffered from a modest impairment of total growth (0.02% area under curve = 11.1 [10.7; 11.5]) (Fig. 1). We confirmed the lethality of 0.2% mp to Pe in liquid culture in a separate experiment, where again no colonies were recovered after 24 h of incubation (Fig. S1). This demonstrates a bactericidal effect of mp on Pe,

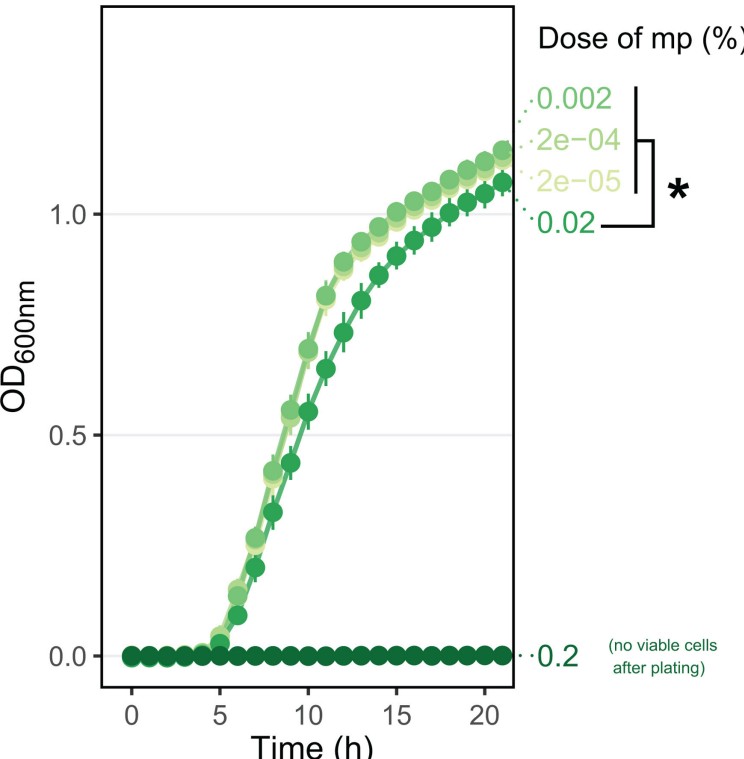

**FIG 1** Effect of mp on Pe growth *in vitro*. Dots with error bars represent the average absorbance at $OD_{600}$ of $N = 8$ replicated wells, with 95% confidence intervals. The concentration of mp is indicated by the green color gradient and by the value (in %) at the tip of the curves. Each well was plated on LB agar after 22 h of growth to check for cell viability. Only the 0.2% mp condition showed no growth, with a lower limit of detection at 333 CFU per mL. The "*" indicates non-overlapping confidence intervals of total area under curve, a proxy for total growth.

at a dose routinely used in *Drosophila* food recipes. Similarly to mp, 0.5% propionic acid (another preservative agent sometimes added to Drosophila medium) also showed strong bactericidal properties toward Pe (Fig. S1B).

## Flies die from low-dose Pe infections in the absence of methylparaben

Given its bactericidal effect in liquid culture, we explored the consequences of dietary mp for the outcome of oral Pe infection at a range of doses ($OD_{600}$ 0.0001–50) in adult *Drosophila melanogaster* flies. Matching previous studies (e.g., 1), in the presence of mp, only the highest dose (100 µL of Pe suspension at $OD_{600} = 50$ per vial) led to severe mortality within a week of infection (Fig. 2A, top row; Δ survival$_{(control − OD50)}$ on the last day of sampling = −0.83 [−0.71; −0.94]; here and after, we give all results as the difference (Δ) of a posterior mean compared to a control condition and with [95% highest posterior density intervals]). Almost no deaths were observed among flies infected with lower doses in the $OD_{600}$ 0.0001–1 range (all Δ survival overlap 0, see Table S3). In contrast, all pathogen doses we used were highly lethal for flies maintained on food without mp (Fig. 2A, bottom row; Table S3). Even the lowest concentration ($OD_{600} = 0.0001$) led to significant mortality within a week, although somewhat lower than the larger doses (Δ survival$_{(control − OD0.0001)}$ on the last day of sampling = −0.38 [−0.19; −0.55]). All these observations were confirmed in an independent replicate experiment, except that we observed no mortality at the lowest Pe dose ($OD_{600} = 0.0001$), most likely due to a generally lower virulence of Pe in this replicate (Fig. S2). Such variation from experiment to experiment in overall Pe virulence is often observed (27).

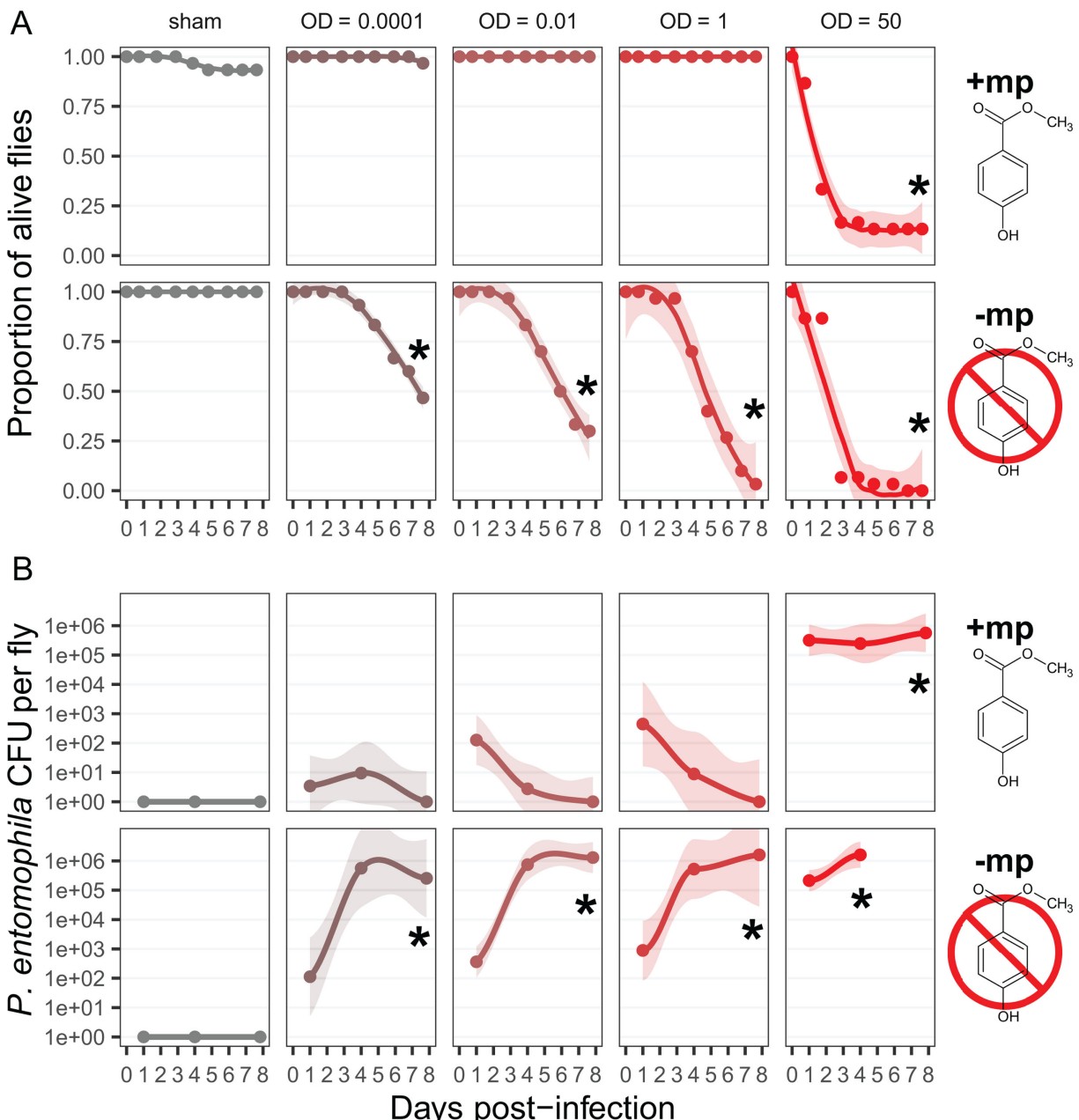

**FIG 2** Survival (A) and pathogen load (B) of female flies exposed to different doses of pathogen (in columns), and in the presence or absence of 0.2% mp in the diet (in rows). The color gradient represents the increasing Pe dose, from gray (sham infection) to red (highest infection dose of $OD_{600}$ 50). In (A), each dot is the average survival proportion of $N = 2$ replicated vials, with 15 flies each. The line represents a loess regression on non-averaged proportions, and the shaded ribbon the 95% confidence interval on this regression. In (B), each dot represents the average number of Pe CFU per fly, out of two flies sampled from $N = 3$ vials. The line represents a loess regression on non-averaged CFU, and the shaded ribbon the 95% confidence interval on this regression. The limits of detection ranged from 40 to $3.2 \times 10^6$ CFU per fly. Missing points are cases where all flies died before the measurement. In all plots, the "*" indicates non-overlapping credible intervals on the last day of sampling from the posterior distribution, compared to the sham-infected condition.

The pathogen load well matched the survival data, with flies showing noticeable mortality only when their Pe load was high (Fig. 2B). In the presence of mp, Pe CFU counts were different from the sham control only at $OD_{600}$ 50 ($\Delta CFU_{(control - OD50)}$ on the last day of sampling = 4.2e + 05 [3.7e + 02; 1.5e + 06]), but in the absence of mp, all infection doses resulted in high loads of $10^5$–$10^6$ CFU per fly within 4 days of infection (Table S4A). In the absence of mp, the median time to death was roughly linear with the log OD of the initial dose (Fig. S3).

## Daily transfer improves fly survival to Pe

The above results imply that, in the absence of mp, Pe multiplies even if initially at a very low dose. This multiplication may be happening both in the fly gut and in the fly food medium. To test for the importance of Pe multiplication in the fly food medium, we evaluated the consequences of a daily transfer of infected flies to new vials with fresh food, thus only allowing Pe present in fly guts and on their body surface to persist through the transfers. This daily change for fresh vial rescued flies' survival, even in the absence of mp (Fig. 3). By resetting Pe growth every day and preventing it from thriving on fly food, we managed to mitigate Pe-induced mortality at lower doses (Δ survival$_{(control - OD0.0001)}$ and Δ survival$_{(control - OD0.01)}$ both overlap 0; see Table S3), but observed a mild mortality at $OD_{600}$ 1 (Δ survival$_{(control - OD1)}$ = −0.16 [−0.04; −0.28]) and a strong mortality only at $OD_{600}$ 50 (Δ survival$_{(control - OD50)}$ = −0.94 [−0.88; −0.99]). Although average CFU quantification visually showed a two to three orders of magnitude reduction in load with transfer, we could not confirm any reduction in individual contrasts with our model (all Δ CFU$_{(transfer - no transfer)}$ overlap 0, see Table S4B). Overall, these results suggest that the lethal effects of Pe are magnified by reinfection with bacteria that multiply in the food medium.

We further strengthened evidence for this hypothesis by allowing five flies infected with a low inoculum of Pe at $OD_{600}$ 0.01 to contaminate their environment in new food vials for 20 h. After this time, we recovered around $10^3$ CFU per vial (Fig. S4). To verify that the Pe continues to grow even in the absence of flies, we removed them and maintained the vials for three more days, at which point the Pe load grew up to $10^6$ CFU per vial.

## High transmissibility of Pe in the absence of methylparaben

In the daily transfer experiment described above, the females were expected to lay eggs in each vial. By keeping and observing these vials after the flies had been removed, we could explore the consequences of parental infection for offspring fitness. Because the number of eggs laid in each vial was unknown, we could not precisely estimate survival. However, the outcome for the offspring was bimodal (Fig. 4A). In the presence of mp, all vials had numerous pupae except in the $OD_{600}$ 50 condition, which showed significant but variable mortality. In the absence of mp, all parental Pe doses led to the almost complete extinction of the next generation. We therefore binarily scored viability at the vial level, as viable or not (Fig. 4B; for precise criteria, see Materials and Methods). Both the presence of mp and the Pe dose applied to parents affected thus defined vial viability ($\chi^2_{(df = 1, N = 6)}$ = 24.6, $P < 0.001$ and $\chi^2_{(df = 1, N = 6)}$ = 22.7, $P < 0.001$, respectively).

Rather than resulting from parent-offspring pathogen transmission, this impaired vial viability might have been caused by infected parents laying non-viable eggs. To exclude this explanation, in a separate experiment, we tested for the consequence of environmental transmission of Pe from non-parental adults to larvae hatched from eggs laid by non-infected parents (Fig. S5). These larvae showed no pupation success when placed on a −mp diet contaminated by infected adults, while their pupation success was comparable to sham conditions on +mp diet (Δ pupation success%$_{(+mp\ infected - -mp\ infected)}$ = 86 [80; 92]). This is consistent with the offspring dying from their own infection rather than from transgenerational costs of parental infection. Of note, we observed that −mp diets also reduced somewhat the viability of sham-infected eggs (Δ pupation success%$_{(+mp\ sham - -mp\ sham)}$ = 25 [16; 34]), possibly resulting from growth of ambient microorganisms that are normally controlled by mp.

We also tested for indirect adult-to-adult transmission of Pe via contaminated food media. We observed that 24 h were enough for a single male infected with a low Pe dose ($OD_{600}$ 0.01) to contaminate their environment and trigger a lethal infection in around 40% of the newly arrived females within a week of exposure (Fig. 5A; $\chi^2_{(df = 1, N = 160)}$ = 17.7, $P < 0.001$). The Pe load of these secondarily infected flies was highly variable, with some completely uninfected, and some with infections over $10^7$ CFU per fly (Fig. 5B). However, this variation in the infection status was well correlated to the survival within a given vial, showing that flies died more in vials with larger Pe abundance (Fig. 5C).

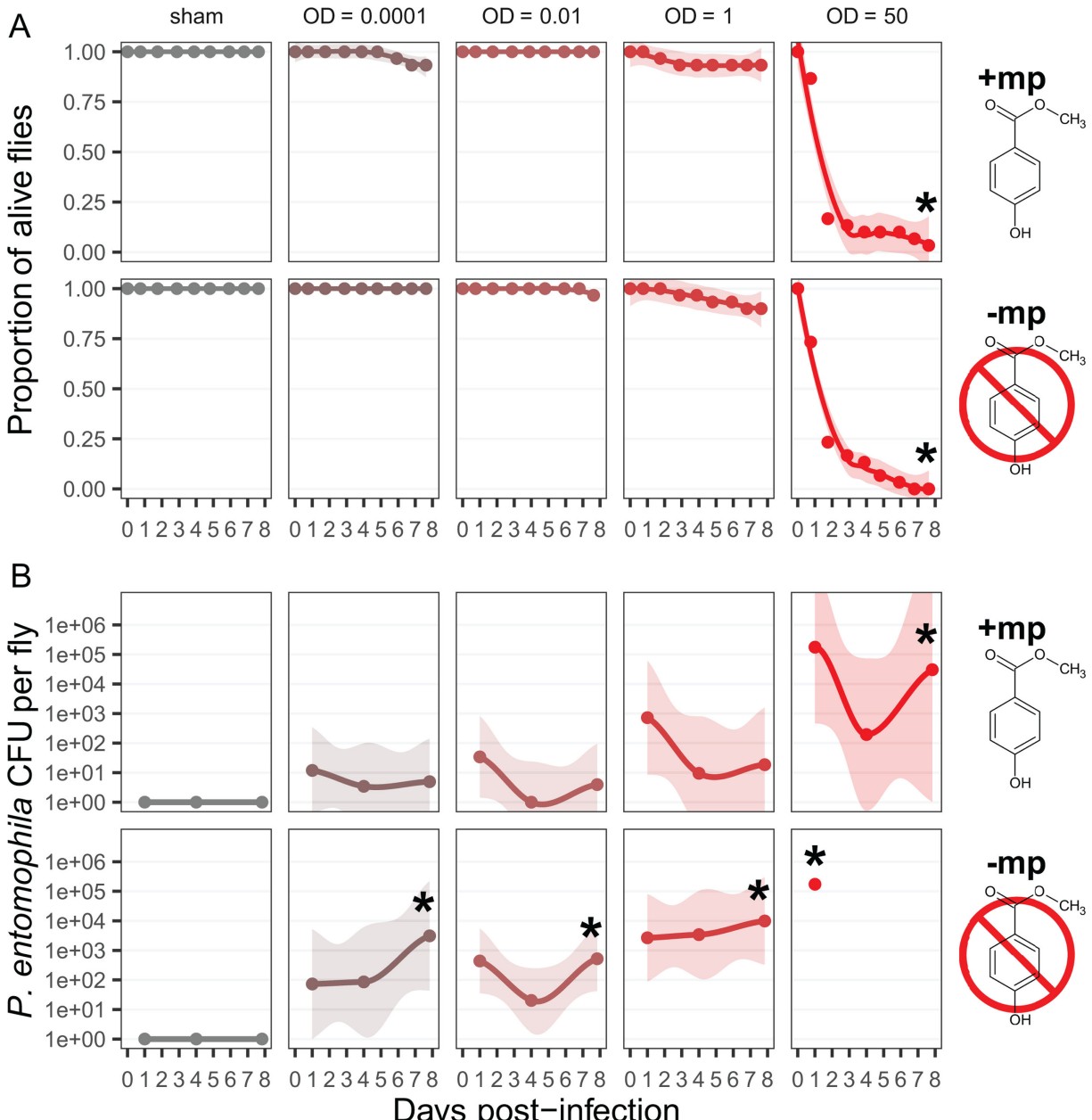

**FIG 3** Survival (A) and pathogen load (B) of female flies daily transferred to fresh vials post-infection, exposed to different doses of pathogen (in columns), and in the presence or absence of 0.2% mp in the food (in rows). See Fig. 2 for detailed explanations on replication, colors, and symbols.

## DISCUSSION

In this study, we tested the effect of methylparaben (mp), a commonly used antifungal preservative added to laboratory *Drosophila* diets, on the consequences of oral infection with the model pathogen *Pseudomonas entomophila* (Pe). We observed strong antimicrobial properties of mp against Pe, limiting both pathogen load and mortality in infected flies maintained on +mp diet. As a consequence, only an extremely high dose of $OD_{600}$ 50, similar to those used in most studies, resulted in significant mortality on a diet containing mp at a standard concentration typically used in *Drosophila* research (0.2%). We showed that this standard concentration of mp exceeds the minimum bactericidal concentration for Pe, resulting in complete lethality within 24 h in liquid culture.

In light of this finding, it is perhaps surprising that extremely high doses of Pe do induce fly mortality on +mp medium. As has been extensively studied, Pe kills the host

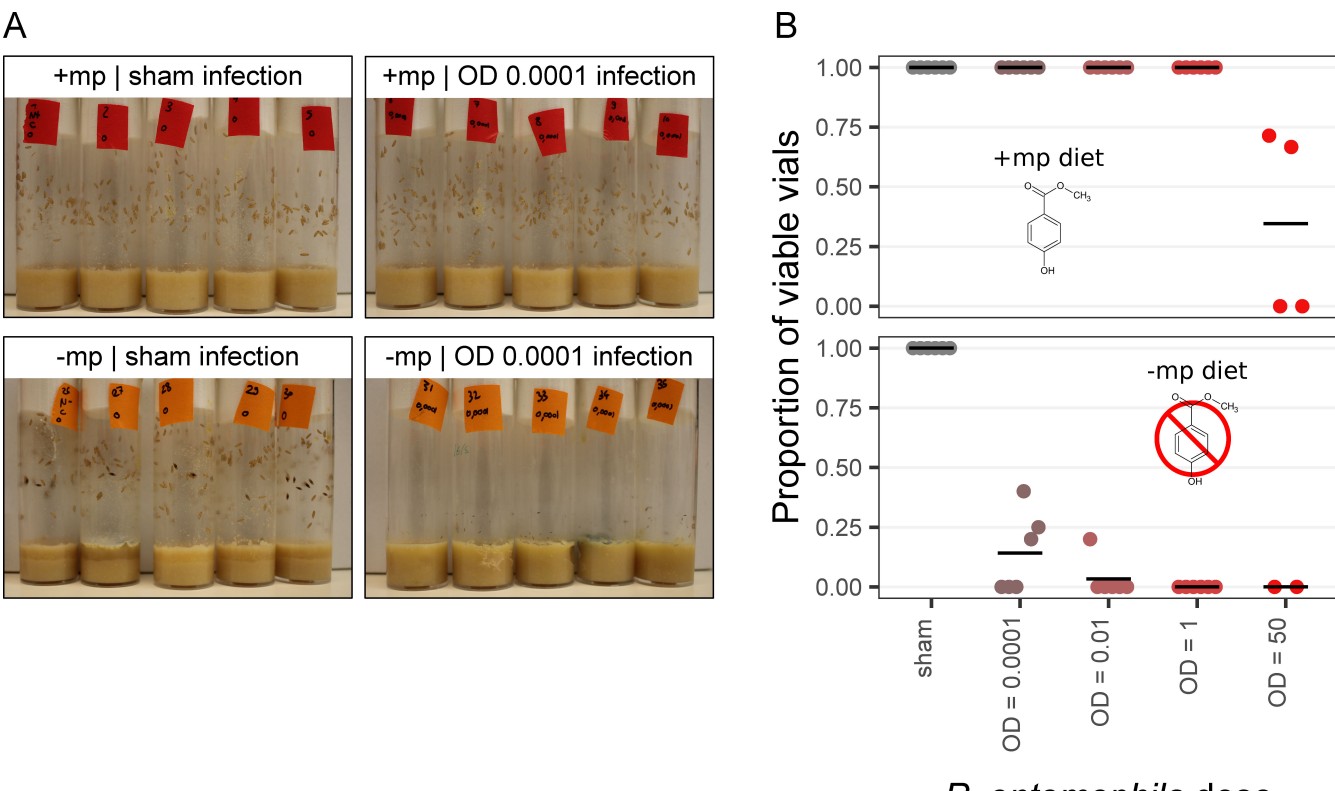

**FIG 4** Photos of offspring vials illustrating the largely binary outcome of egg-to-pupae viability (A) and proportion of viable offspring vials (B) depending on the presence or absence of 0.2% mp in the food. In (A), the four pictures show representative vials exposed to sham or OD 0.0001 infections, in the presence or absence of 0.2% mp in the food. Vials were obtained after the first of the daily fly transfers, and pictures were made 7 days after that transfer. Dead early instar larvae are visible only in the −mp OD 0.0001 condition, with no pupae. In (B), the color gradient represents the increasing Pe dose, from gray (sham infection) to red (highest infection dose of $OD_{600}$ 50). Each dot represents the proportion of viable vials at one vial change timepoint, for $N = 6$ time points. The proportion was calculated as the number of viable vials out of the total number of considered vials for a given timepoint and treatment (five vials when all considered). Vials were considered viable if >5 pupae and <5 dead larvae were visible; vials were considered not viable if <5 pupae and >5 dead larvae were visible; other vials were not considered for the analysis.

by compromising gut homeostasis and integrity through action of bacterial toxins and inhibition of epithelial renewal, possibly exacerbated by ROS produced by the gut as a part of the immune response (2, 7, 14, 28). It is possible that at these very high doses, even weakened or dying Pe cells induce sufficient harm either directly (e.g., by releasing toxins) or via inducing harmful immune response. Consistent with this view, high mortality in flies can be induced by repeated ingestion of large amounts of heat-killed Pe (29). Nonetheless, in our experiment, flies infected with the highest dose retained a stable load of live Pe even on +mp medium. Persistence of live Pe over several days following infection with massive doses on +mp media had been reported before (30), although other studies reported gradual clearance of the pathogen under such conditions (27, 28). Possibly, this persistence of Pe despite mp presence might be a manifestation of the "inoculum effect," a phenomenon of antibiotic resistance appearing when exposing massive cell densities to antibiotics (31). While this phenomenon is easily understood when bacteria can degrade the antibiotic compound (e.g., β-lactamases for β-lactam antibiotics), other mechanisms that could be at play with antimicrobials such as mp have not been clearly established (32, 33). We also note that low levels of Pe—at the limit of the detection threshold—persisted in flies infected with lower doses on +mp medium. While we do not have evidence for this, the most plausible explanation is the existence of mp-free "refugia," e.g., on the walls of culture vials or on the foam plug. However, these loads are on the order of 1–10 CFUs per fly, i.e., several orders of

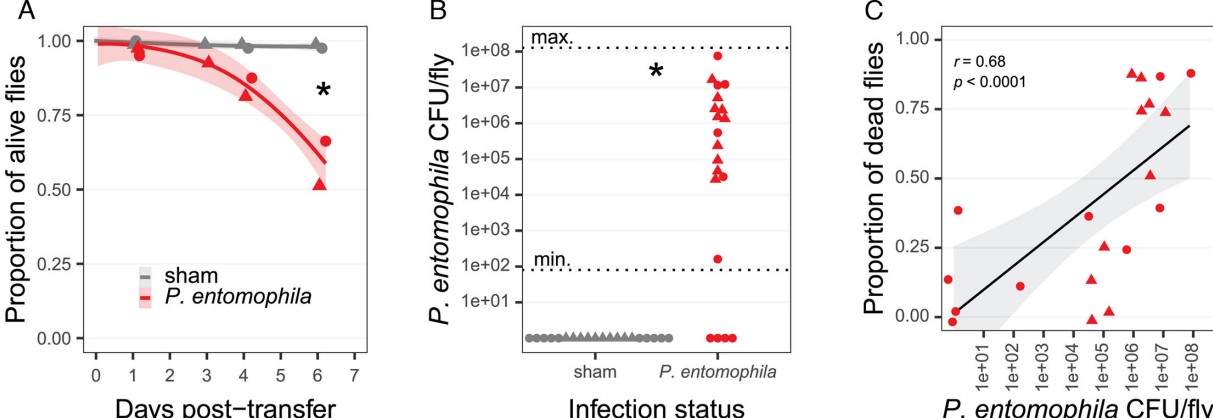

**FIG 5** Survival (A), Pe load (B), and correlation between survival and Pe load (C), in the case of an indirect adult-to-adult transmission of Pe infection. Single sham-infected or Pe-infected males were transferred in a vial containing −mp diet, removed after 24 h, and replaced with eight females on which we measured survival and Pe load (see Materials and Methods for details). Results for tests on +mp diet are not included here as no mortality and no Pe CFU were observed, both in sham-infected and Pe-infected treatments. In (A), each dot is the average survival proportion of $N = 20$ replicated vials, with eight females each. The line represents a loess regression on non-averaged proportions, and the shaded ribbon the 95% confidence interval on this regression. The "*" indicates a significant survival difference on the last day of sampling ($X^2_{(df = 1, N = 160)} = 17.7$, $P < 0.001$). In (B), each dot is the CFU count on the last day of sampling (6 days) of a single fly sampled from a single replicate ($N = 20$ replicates). The "*" indicates a significant CFU difference on the last day of sampling (Mann-Whitney U, $P = 0.005$). In (C), dots combine the proportion of dead flies in a vial (y axis) with the Pe load from a fly sampled in the same vial (x axis), only keeping the Pe-infected condition. For (B) and (C), the detection range, indicated with the horizontal dotted lines in B, was 80–$1.2 \times 10^8$ CFU per fly. For all plots, colors represent the infection treatment of the males that preceded the females in the vials, with gray indicating sham-infected males and red Pe-infected males with $OD_{600}$ 0.01. In all plots, circles and triangles represent independent experimental blocks.

magnitude below the load associated with mortality. These observations should be treated with caution, as the CFU measurements were based on a single experiment and were not independently replicated.

We demonstrated that in the absence of mp, Pe was lethal at doses six orders of magnitude lower than those lethal in the presence of mp and typically used in previous studies (e.g., 1). In the absence of mp, Pe cells reached loads of about $10^6$ CFU per fly regardless of the initial infection dose, including the smallest ones. This implies that Pe could proliferate unchecked, eventually reaching a critical lethal load. In line with this interpretation, median time to death declined linearly with the logarithm of initial dose; such a linear relationship is expected if mortality starts once an exponentially growing pathogen population reaches a critical threshold. Furthermore, the onset of mortality coincided with the load of Pe reaching *ca*. $10^5$ CFU per fly. While this correlation may suggest that $10^5$ CFU per fly is the lethal load, the actual tipping point beyond which the host is doomed may be occurring at a lower concentration, with the collapse of immunity and loss of gut homeostasis allowing further pathogen proliferation (7, 28, 34). Irrespective of the degree to which pathogen growth is modulated by flies' immune response, it is clear that in the absence of mp, fly immunity ultimately fails to prevent mortality under typical conditions of *Drosophila* lab culture.

The capacity of Pe to reach critical densities seemed key for observing fly mortality. Even in the absence of mp, we found that pathogen load following Pe infection is reduced and mortality largely eliminated (except at the highest dose) by daily transfer of infected flies to new uncontaminated food sources. Such transfer can be seen as simulating natural movements of flies to new food patches. This result indicates that Pe does not multiply sufficiently in fly guts to reach lethal loads; growth in the external environment of fly medium—and presumably in nature on substrates on which flies feed —is necessary. Thus, adult flies can escape mortality even when already infected, if they detect and avoid further exposure to Pe contaminated patches, behavior that has been reported (35). Yet, these infected adults still suffer loss of Darwinian fitness—we have shown that they can contaminate new food patches and thus transmit the pathogen to

their offspring and other larvae that happen to live in a patch visited by infected flies. This is highly lethal to the larvae who, in contrast to adults, have a reduced ability to move to new uncontaminated food patches. We also demonstrated indirect transmission from infected adults to other adults or larvae via contaminated environment, showing that no social physical interaction was required for transmission in the absence of mp. This capacity to transmit the pathogen, at least to larvae, was greatly reduced by mp, likely by a combination of reduced pathogen shedding by the infected individuals and inhibition of its growth on the contaminated medium.

These results are consistent with the prediction that indirect transmission through the environment is often better suited for highly virulent pathogens in non-eusocial insects (36–38). The combination of low initial Pe concentration combined with the indirect transmission through a single infected individual likely created conditions conducive to demographic stochasticity, explaining the observed variability in survival and Pe load (39). Even though Pe transmission success was imperfect, we still observed clear mortality that was directly correlated with Pe load on the last day of sampling.

Overall, our findings suggest that the pathogenic and epidemiological potential of Pe in *Drosophila* has been historically underestimated due to the nearly universal presence of mp in artificial fly diets, which compelled researchers to use biologically unrealistic pathogen concentrations. In the absence of such additives, Pe is not only deadlier than expected at lower—and presumably more ecologically realistic—doses, but also transmissible between individuals. This pathogen exhibits characteristics—specifically, low-dose virulence and high transmissibility—that make it suitable for exploring ecological questions, such as population-level bacterial dynamics in natural or semi-natural experimental setups. The lethality and transmissibility of Pe both under laboratory conditions and in nature are likely to be affected by numerous factors, such as the composition of the diet and the composition and abundance of microbial community (40). Nutrient availability may limit growth of Pe, and it may be further inhibited by reduction of pH by fermenting bacteria, as has been reported (30). The pathogen infection and transmission dynamics may also be affected by host sex (8); for practical reasons, our study only measured mortality in adult females and used males to trigger indirect infections. These limitations notwithstanding, our study gives support to the notion that Pe has the potential to be an ecologically relevant natural pathogen of *Drosophila*, as it has been proposed by its discoverers (1).

While we focused on Pe as the pathogen and mainly on mp as the antifungal agent, it is likely that our results are relevant for other pathogens and other preservatives. Like mp, our preliminary experiment indicated that propionic acid, another widely used preservative in artificial fly diet, is also able to kill all Pe cells after 24 h of exposure *in vitro*. Interestingly, the only study that used mp-free fly food but still needed high doses of Pe for oral infections included propionic acid in the food recipe (41). Hence, multiple antifungal preservatives could lead to similar protection against bacterial pathogens, thanks to incidental antibacterial properties. Being broad-spectrum antimicrobial agents, food preservatives could also affect other species besides Pe. Indeed, inhibitory effects of mp have been observed in several medically-relevant pathogen species (15, 16, 42), and both *Pseudomonas aeruginosa* and *Serratia marcescens* require large oral infections to kill flies maintained on a diet containing preservatives (43, 44).

To conclude, we found that the use of antifungal preservatives in artificial *Drosophila* diets resulted in underestimation of Pe virulence and transmissibility compared to what is probably happening in natural conditions. Given the widespread use of such preservatives in industrial processed food, it is not unlikely that these compounds have effects beyond Pe and *Drosophila*, potentially influencing a wide range of bacterial infections across biological models. Taking these effects into account is essential for accurate interpretation of experimental results, for reproducibility, and more generally for understanding the natural dynamics of host-pathogen interactions.

## MATERIALS AND METHODS

### Fly maintenance

For all experiments, we used a *Wolbachia*-free wild-type fly population collected in 2007 in the Valais, Switzerland. Stocks were kept in outbred conditions in a thermoregulated room set at 25°C ± 0.5°C, 60% RH, and 12L:12D light cycle. The standard diet of flies was composed of brewer's yeast (2% wt/vol), cornmeal (5.2% wt/vol), sucrose (11% wt/vol), agar (0.8% wt/vol), and methylparaben (ref H5501, Sigma-Aldrich, USA) (1.1% vol/vol of 20% stock solution in pure ethanol) mixed in water. The concentration of mp was in line with standard food recipes found elsewhere (see references from Table S1). In the manuscript, "+mp diet" refers to the standard diet, and "−mp diet" refers to a modified standard diet without mp. We performed all fly transfers with brief $CO_2$ anesthesia.

### Bacterial cultures

For all experiments, we used the L48$^T$ strain of *Pseudomonas entomophila* (kindly shared by Bruno Lemaitre), grown overnight in LB medium (ref 240230, BD Difco, USA) at 30°C, 150 rpm. When required, we standardized the concentration to the desired $OD_{600}$ by pelleting the cultures at 3,000 rpm for 5 min, discarding the supernatant, and resuspending in a sterile solution of 5% sucrose for infections or sterile 25% glycerol solution in PBS buffer for plating and storage. For each experiment, we plated Pe to verify that $OD_{600}$ 1 corresponded approximately to $10^9$ CFU/mL.

### Effect of methylparaben and propionic acid on Pe *in vitro*

We tested MIC and MBC of methylparaben on Pe *in vitro*. In a 96-well microplate, we grew a suspension of Pe (starting $OD_{600}$ 0.001) in LB medium supplemented with mp at five different doses, ranging from 0.2% to 0.00002%. We made eight replicated wells per mp dose and incubated the microplate at 30°C with 300 rpm agitation, in a spectrophotometer (HIDEX sense, Finland) with hourly $OD_{600}$ measurements. After 22 h of incubation, we plated each well on LB agar. We checked for the presence of colonies after 20 h of incubation at room temperature.

### Oral infection procedure

We largely followed standard infection procedures, extensively described by Siva-Jothy et al. (10). Before the infection, we starved flies for 4 h in tubes containing water agar (1%), with 8–15 flies per tube. We prepared the infection vials by adding a filter paper on the surface of a new water-agar vial and pipetted down 100 µL of the pathogen suspension in 5% sucrose with the desired $OD_{600}$ or 100 µL of the 5% sucrose solution only (sham control). We transferred the flies to these vials and, after 20 h of infection, transferred them again to fresh vials containing food with or without mp. We then monitored the flies maintained at 25°C ± 0.5°C, 60% RH, and 12L:12D light cycle for up to 8 days to record survival and Pe load. We recorded survival with daily counting of dead individuals. We measured the bacterial load in flies by plating alive individuals sampled from separate dedicated vials.

### Gut load of *Pseudomonas entomophila*

To measure the Pe load in flies' guts, we sampled 1–2 random alive individuals per vial at different time points post-infection, depending on the experiment. We briefly dipped the collected flies in 70% ethanol, vortexed, discarded the ethanol, and let the flies dry for 30 min to ensure killing external bacteria. We then crushed the flies in 400 µL of 1:1 mix of sterile 40% glycerol and sterile PBS, with a 2 mm steel bead using a tissue homogenizer (Precellys evolution, Bertin, France) at 4,500 rpm for 1 min. After a serial dilution range, we plated 5 µL of each dilution on *Pseudomonas* isolation agar medium (ref17208, Merck, Germany), and counted the colony-forming units (CFU) after 20 h of growth at room temperature. The CFU detection range spanned from 20–80 CFU per

fly (= 1 colony in the undiluted homogenate) up to $1.6 \times 10^6$–$1.2 \times 10^8$ CFU per fly (= 200 colonies in the most diluted homogenate), depending on the experimental block. Colonies counted this way likely corresponded to Pe cells from the gut, since the leakage of Pe from the gut into the hemolymph is associated with imminent death of the flies (28). Yet, we cannot exclude that for some samples, counts also include bacteria located elsewhere in the flies.

## Effect of methylparaben and daily transfer on fly survival

Six-day-old individuals were sexed under $CO_2$ anesthesia, keeping only females (which were likely mated, having spent several days in mixed sex groups). We then divided females into multiple vials (15 females per vial) of +mp diet or −mp diet. After 3 days on this new diet, we proceeded to infections (see "Infection procedure" section for details). We used five concentrations of Pe ($OD_{600}$ = 50, 1, 0.01, 0.0001, or 0), and after 20 h of infection, we transferred the flies to fresh vials containing the +mp or the −mp diet. Then, we maintained the flies either in the same vial for up to 7 days or transferred them daily to new vials with fresh food, recording survival rates or Pe load. This daily transfer condition allowed us to test whether Pe was stably colonizing the gut or whether it required constant replenishment via feeding, as observed with most bacteria from the fly microbiome (40, 45). In the condition with daily transfer, we always counted the dead flies and sampled individuals for Pe load measurement just before the next transfer. Dead individuals were counted daily. They were not removed from the vial they died in, meaning they were not transferred to the new vial in the daily transfer condition. In total, we tested 20 different conditions (2 +mp/−mp diets × 2 transfer/no transfer × 5 infection doses). We used two replicate vials per condition for survival and three replicate vials for Pe load measurement. Two independent experimental replicates were performed, with minor differences explained in Fig. S2.

## Adult-offspring pathogen transmission

We kept the used vials from the daily transfer condition (see above) to assess the pupation success of the offspring, by observing dead larvae and pupae 6 days after the change ($N$ = 5 replicated vials per condition and per day). The variable number of individuals caused by death events strongly affected the number of eggs laid. For that reason, we opted for an arbitrary binary classification of the offspring vials: a vial was considered as viable when the number of dead larvae was below five and the number of pupae exceeded five, and a vial was considered as non-viable when the number of dead larvae exceeded five and the number of pupae was below five. All vials not matching these criteria (21/300) were discarded from the analysis, as they likely contained no offspring to start with.

## Indirect adult-offspring and adult-adult pathogen transmission

To investigate the extent of Pe pathogenicity at low doses and differentiate between environmental and social transmission routes, we conducted a separate experiment testing indirect pathogen transmission from adult flies to offspring and other adults. Four-day-old individuals from the stock population were sexed under $CO_2$ anesthesia. We then divided individuals into multiple single-sex vials (8 females or 8 males per vial) of −mp standard diet. After 3 days on this new diet, we started the infection procedure on males (see "Infection procedure" section for details). We used two infection treatments (Pe at $OD_{600}$ 0.01 and sham control). For adult-offspring transmission, we infected males for 6 h and transferred groups of five into fresh vials containing −mp or +mp standard diet; for adult-adult transmission, we used a slightly different setup as we infected males for 20 h and transferred single males into fresh vials containing −mp or +mp standard diet. After 24 h, we discarded the males and introduced 20 uninfected eggs or eight uninfected females into each used vial. We observed the vials for 6 days, recording pupation for the vials in which we added eggs (six vials per condition), and survival

for the vials in which we added females (two experimental blocks with 10 replicates per condition each). We also plated surviving females at the end of the experiment to quantify Pe load.

## Environmental load of *Pseudomonas entomophila*

We measured the Pe load in fly food, testing whether infected flies could contaminate their environment with Pe, and whether Pe could survive and grow depending on the presence of mp. We infected groups of five males or five females using the infection protocol described earlier, either with a suspension of Pe at $OD_{600}$ 0.01 or with a sham control infection. After 6 h in the infection vials, we transferred flies into fresh vials containing −mp or +mp diet and let them use those vials for 20 h before discarding them, keeping only the empty vials. We performed CFU measurements at two time points, right after discarding the flies, and 68 h after discarding the flies ($N = 3$ replicated vials per treatment combination and per time point). To collect samples, we pipetted 1 mL of PBS in the vials, vortexed for 10 s to suspend potential biofilms, and used this suspension for plating. After a serial dilution range, we plated 3 µL of each dilution on *Pseudomonas* isolation agar medium (ref17208, Merck, Germany), and counted the CFU after 20 h of growth at room temperature. The CFU detection range spanned from 333 CFU per vial (= 1 colony in the undiluted homogenate) up to $5.3 \times 10^8$ CFU per vial (= 200 colonies in the most diluted homogenate).

## Statistical analysis

We analyzed most data sets in a Bayesian framework, using R version 4.2.1 (46) and the brms package (47) as frontends for the Stan language (48). We used the tidyverse, tidybayes, bayesplot, and patchwork packages for data preparation, model evaluation, and plotting (49–52).

We analyzed flies' survival after a direct Pe infection using a Bayesian binomial linear model with a logit link function, using a data set of the survival on the last day of sampling only (t = 183 h). Our model included mp presence, infection dose measured as $OD_{600}$, and vial change as fixed factors, with all interactions.

We analyzed Pe CFU load after a direct Pe infection using a Bayesian linear model, using a data set of the CFU on the last day of sampling only (mostly t = 188 h, or earlier datapoints for a few vials that had no more survivors at 188 h). Our model included mp presence, infection dose measured as $OD_{600}$, and daily transfer as fixed factors, with all interactions.

We analyzed pupation success after indirect Pe infection using a Bayesian linear model. Our model included mp presence and infection of the vials as fixed factors, plus their interaction. We also included a group-level random effect associated with each replicate.

We analyzed Pe CFU in vials used by flies infected or not with Pe using a Bayesian linear model. Our model included mp presence, infection with Pe, and time point as fixed factors, plus their interaction. We also included a random effect associated with each sex.

We fit our models using weakly informative priors inspired by McElreath (53): Normal (0, 1.5) prior for the intercept, Normal (0, 1) prior for the random effect, and Normal (0, 1) or Normal (0, 5) priors for the fixed effects in the survival model and the CFU models, respectively. We ran four chains for 10,000 iterations, with the first half of each chain used as a warmup. We give all results as the difference (Δ) of a posterior mean [95% highest posterior density intervals] compared to a control condition. By construction, our Bayesian approach did not require any correction for multiple comparisons (54).

We analyzed the data from *in vitro* Pe growth and Pe transmission experiments (except indirect adult-offspring transmission) in a frequentist framework using R.

We analyzed *in vitro* Pe growth using a linear model of the area under the curve calculated with growthcurver (55), with mp concentration as the fixed factor.

For the direct transmission to the next generation, we analyzed the pupation success using the mixed binomial generalized linear model with a logit link function from the

lme4 package (56). We included mp and Pe dose as interacting fixed effects, and the day of change as a random effect. For survival on the last day of sampling in the adult-adult transmission experiment, we used a mixed binomial generalized linear model with a logit link function. We included the infection status as a fixed factor, the experimental block as a random effect, and the vial identifier as an observation-level random effect to control for overdispersion (57). For CFU, we compared the two distributions (infected vs sham) with a Mann-Whitney U test. Finally, we checked the correlation between survival and Pe load using Pearson's correlation test.

## ACKNOWLEDGMENTS

We warmly thank the members of the Kawecki group who all encouraged us to pursue this work initially designed as a technical check.

This work was supported as a part of NCCR Microbiomes, a National Centre of Competence in Research, funded by the Swiss National Science Foundation (grant number 51NF40_180575), and the Swiss National Science Foundation (grant 310030_184791 to T.J.K.).

Y.H.: conceptualization, investigation, formal analysis, writing—original draft, writing—review and editing; B.C.-D.: investigation, writing—review and editing; J.G.: conceptualization, investigation; C.L.M.: investigation; T.J.K.: conceptualization, writing—review and editing, supervision.

## AUTHOR AFFILIATION

[1]Department of Ecology and Evolution, UNIL - University of Lausanne, Lausanne, Switzerland

## AUTHOR ORCIDs

Youn Henry  http://orcid.org/0000-0001-5972-9136
Berta Canal-Domènech  https://orcid.org/0000-0003-1447-5704
Jaime González  https://orcid.org/0000-0002-8337-0941
Tadeusz J. Kawecki  http://orcid.org/0000-0002-9244-1991

## FUNDING

| Funder | Grant(s) | Author(s) |
| --- | --- | --- |
| National Center for Competence in Research Microbiomes (NCCR Microbiomes) | 51NF40_180575 | Tadeusz J. Kawecki |
| Swiss National Science Foundation | 310030_184791 | Tadeusz J. Kawecki |

## AUTHOR CONTRIBUTIONS

Youn Henry, Conceptualization, Formal analysis, Investigation, Writing – original draft, Writing – review and editing | Berta Canal-Domènech, Writing – review and editing | Jaime González, Investigation, Writing – review and editing | Christine La Mendola, Investigation | Tadeusz J. Kawecki, Conceptualization, Supervision, Writing – review and editing

## DATA AVAILABILITY

Data sets and scripts are available on the Zenodo repository DOI 10.5281/zenodo.15115312.

## ADDITIONAL FILES

The following material is available online.

## Supplemental Material

**Supplemental material (Spectrum03065-24-s0001.pdf).** Tables S1 to S4; Fig. S1 to S5.

## Open Peer Review

**PEER REVIEW HISTORY (review-history.pdf).** An accounting of the reviewer comments and feedback.

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
