## [Reviewer comments · Microbiology Spectrum]

Microbiology Spectrum

Standard ingredient of *Drosophila* medium reduces transmission and virulence of the gut pathogen *Pseudomonas entomophila*

Youn Henry, Berta Canal-Domènech, Jaime González, Christine La Mendola, and Tadeusz Kawecki

Corresponding Author(s): Youn Henry, University of Lausanne

Review Timeline:

Submission Date:	November 26, 2024
Editorial Decision:	January 20, 2025
Revision Received:	April 4, 2025
Editorial Decision:	April 18, 2025
Revision Received:	April 24, 2025
Editorial Decision:	June 2, 2025
Revision Received:	July 4, 2025
Accepted:	July 8, 2025

Editor: John Chaston

Reviewer(s): The reviewers have opted to remain anonymous.

Transaction Report:

DOI: <https://doi.org/10.1128/spectrum.03065-24>

Re: Spectrum03065-24 (Standard ingredient of *Drosophila* medium reduces transmission and virulence of the gut pathogen *Pseudomonas entomophila*)

Dear Dr. Youn Henry:

Thank you for the privilege of reviewing your work. Below you will find my comments, instructions from the Spectrum editorial office, and the reviewer comments.

You will see that both reviewers valued the contributions of the work, but both also had some concerns about the limitations of the study and, in particular, that the discussion was over interpreted. You will see this in the comments below, but it also came up in their confidential comments to me directly. If you choose to revise I encourage you to carefully consider their comments, and to make sure that the discussion addresses other interpretations and makes it clear when speculation is based on data or may be more like a of evidence you favor for future work. It may not be necessary to remove text reflecting the latter if the caveats and alternatives are properly recognized, but the key is that they should be addressed.

Revision Guidelines

Sincerely,
John Chaston
Editor
Microbiology Spectrum

Reviewer #1 (Comments for the Author):

In this study, Henry et al. ask how the presence of methylparaben (mp) impacts the pathogenicity of oral infections with the Gram-negative bacteria *Pseudomonas entomophila* (Pe) in *Drosophila melanogaster*. Generally, in the field, researchers deliver a significantly high OD (20-200 OD) of Pe to observe infection phenotypes. The authors provide a simple literature review showing that many studies using high inocula of Pe also fed their flies on diets with food preservatives like mp, while the few studies that use a low OD of Pe (less than OD 1) do not use any preservatives. Thus, the authors explore how five different inoculation doses of Pe (uninfected sham, OD 0.0001, OD 0.01, OD 1, and OD 50) impact infection survival of flies fed on diets with or without mp. Interestingly, diets without the preservative increase susceptibility to Pe infection even at lower doses than what is commonly used in the field. This increase in susceptibility to Pe infection in the absence of preservatives is ameliorated when flies are transferred daily to fresh food. Additionally, the authors show some evidence that transmission of Pe is increased when flies are fed preservative free food. This is an important study that interrogates how artifacts of typical fly lab media alter the pathogenicity of Pe. Some additional information would further improve the study.

Major comments

1. Figure 1: The authors show in vitro that 0.2% mp can completely inhibit growth after a 24-hour incubation, which suggests a bactericidal effect of the preservative. Line 121 in the text the authors state a personal observation of a potential bacteriostatic effect of only 6-hour exposure to the preservative. These methods aren't explained in the methods section nor is the motivation for this 6-hour timepoint explained. It is challenging to interpret a complete bactericidal effect based on the single in vitro experiment performed. Have the authors tried inoculating fresh cultures with the overnight cultures grown with methylparaben to see if anything can grow? Including a minimum inhibitory concentration (MIC) assay and a minimum bactericidal concentration (MBC) is critical for definitively supporting the authors claim that mp is bactericidal to Pe. Having a bactericidal and bacteriostatic antibiotic would be helpful positive controls if accessible.
2. Figure 2 and Figure S2: Aligned with the idea of whether mp is bactericidal or bacteriostatic, have the authors tried scoring the CFU counts in the food to see if they similarly match the CFU reflected in the fly? Or whether Pe is detectable in mp food? Understanding what the loads of Pe are in the food provides more certainty of whether mp is bactericidal and provides more context when interpreting the mother-larvae transmission experiments (e.g. the number of cells that larvae maybe exposed to).
3. Figure 3: How do the authors disentangle pathogen transmission from viability of the progeny? Hudson, AL, et al (2023, JEB) show that diets containing ~14% protein (reflecting the protein content in the diet used in this study) lead to less viable progeny post oral infection with *P. aeruginosa* than progeny from flies fed on a high protein diet (~30% protein). They also show that the viability of progeny from infected females decrease over the time of the infection. Potentially, the lack of protein coupled with the stressors that come from carrying a higher bacteria burden in females on preservative free food could impair the viability of their progeny. An additional experiment such as having the eggs from uninfected females laid on vials from infected flies would be another form of evidence to decouple transmission of the pathogen from protein availability for the progeny/mother. Knowing how much CFUs are present in the food coupled with exposing progeny from uninfected mothers to infected vials could help clarify transmission to progeny.
4. Figure 4: I think the evidence that transmission of Pe is higher on mp absent food is weaker here since the authors only show how transmission is possible on the mp absent food. The authors cite Jothy et al (2018) for their oral infection procedure, and they also describe a method for measuring transmission by quantifying the amount of bacterial shedding. Have the authors tried using this method to assess how presence of mp impacts how much Pe infected flies shed?

Minor comments:

1. Figure 1S: Propionic acid has a similar result to the presence of mp where there are no detectable CFUs. What are the justifications for focusing on mp for the study when propionic acid shows to have a similar effect? It could be worthy to repeat some of the survival/CFU load measurements with the OD 1 and OD50 inoculum. This could be another study, so, alternatively, expanding discussion on lines 273-276 on the impact of propionic acid on Pe inoculum to include a need for future experiments.
2. Figure S2B shows that there is a low stable CFU of Pe in flies fed mp when transferred to fresh food, however, Figure 2B shows that there are no detectable loads when flies are not transferred to fresh vials. What is the authors interpretation of mp being bactericidal, however it does not clear the infection when flies are transferred to fresh food vials?
3. Figures 1 & 3: The methods section describes the statistical analyses used for both of these figures, however, the stats for these experiments are not present in the figure or the figure legend for both figures.
4. Line 127-128: For clarity for the reader, stating the dose range will be helpful when stating what their expectations for infection survival are. Can clarify here that "the expectations" are based on observations in the literature which authors have already cited

(lines 66-68), which should be cited here as well when referring to what one expects.

5. Discussion lines 217-218: I think it's a stretch for the authors to state that only bacterial load drives Pe pathogenicity and that no Pe activity impacts survival phenotypes without any examples share of known mechanisms of Pe activity that cause host death/disease. Figure 2 shows that the -mp flies lead to similarly high pathogen loads that correspond to the death of the population. For example, OD1 and OD50 have similar Pe loads at the time the population crashes. Other infections inoculated at high doses like ecc15 do not cause the entire population to die like Pe does at similar loads, which suggests activity of the bacteria leading to high levels of acute lethality.

6. Lines 255-260: I'm not entirely convinced that the paper cited being an example of mp exposure entirely explaining the route of infection differences. The hemocoel and the gut are still two vastly different environments for Pe to be exposed to in the fly even if mp is not present in the hemolymph. For these claims, the authors should clarify their interpretations of what they mean by a "stronger immune responses against Pe" in context of the experiment they are referring to. For example, the authors could share alternative ideas of how lower ROS production to avoid sepsis syndrome requires description of the role of ROS in the gut. Overall, I felt like part of the discussion section makes large interpretations of what maybe occurring in other papers without strong evidence that could suggest otherwise.

7. Line 328: The methods say that 20% methylparaben in 1% ethanol is added to a suspension of Pe in LB. But it does not state that the final concentration of the mixture is 0.2%, which matches the concentration of the fly food. What were the actual volumes used to calculate the final volume of this mixture then 0.2%? Based on the concentration provided, I believe you'd need to perform a 1:100-fold dilution, for example add 10 uL of 20% methylparaben to 990 uL of PE LB culture to achieve a 0.2% final concentration.

8. For the different OD doses, what are the approximate CFUs delivered per fly?

9. For all figure legends, should state what the concentration of mp is in the food.

10. Line 138 (and throughout text): For clarity to the readers, alternatively say "end of the infection time course" or "last day of sampling"?

11. Methods Adult-offspring pathogen transmission: how do the authors determine the larvae are dead (e.g. black/melanized, immobile)? Are these observations only of dead L3 larvae?

12. Line 317: Spell out the entire name of the species

13. Line 364: "Six day old" instead of "Six days old"

14. Line 405: What is the justification for using Bayesian framework for analyzing these datasets?

Reviewer #2 (Comments for the Author):

In this manuscript, Henry et al. demonstrate that the lethality of a *Drosophila* pathogen, *Pseudomonas entomophila*, is strongly affected by the presence or absence of preservatives that are nearly universally used in fly media. In the field of *Drosophila* immunity, *P. entomophila* is a very commonly used pathogen in research, but it is often used at very high concentrations in order to kill flies when infected orally. Here, the authors show that methylparaben (and propionic acid), an antimicrobial preservative that is used in most standard fly food recipes, acts to reduce the among individuals. This study is important to the insect immunity research community lethality of the bacteria. In fly food media prepared without the preservative, the pathogen is lethal at much lower concentrations. The authors also perform experiments that indicate the preservative may reduce transmission of pathogens as these results indicate that ubiquitous ingredients in fly food may inadvertently affect the interpretation of some experimental results and that more ecologically relevant experiments and results may be possible in the lab (ie, with pathogen concentrations at a more natural level) without the preservatives. Overall, the manuscript is straightforward, with some additional input needed on the points below:

- For all of the experiments, the level of replication was somewhat unclear. The number and types of samples within each treatment group and data point were laid out, but it was unclear if full experiments with all treatment conditions were repeated at different times. Were experiments repeated on different days with new biological samples? If experiments were repeated, it would help to state so in methods/captions. If not, it would be important to address or discuss this
- Fig 3 & parent to offspring pathogen transmission experiment:
 - o The scoring of offspring viability was based on the number of visible dead larval and pupal offspring in each vial six days after

adults laid eggs. It is difficult to understand how accurately this could be done as standard fly media is an opaque brown color which would obscure many or most larvae unless they happen to be at the edge of the vial. How were larvae viewed reliably and not missed in any vials and how was it determined if the pupae had died by day 6, when most would still be early in pupal development? Also, why was the value of 5 chosen as the cutoff for viability?

o Line 176-177: The text describes this experiment as measuring egg to pupal survival rate, but it seems that eggs were not counted and the metric used is not really egg to pupal survival, but rather number of visible dead offspring.

o The results show lower offspring viability without the preservative and the authors suggest that this indicates transmission of the pathogen from adult to offspring. However, there is no direct confirmation that the pathogen was passed on to the next generation since offspring were not tested for the pathogen (measuring CFU's etc.), and were only counted for visible survival. The language should be amended to reflect that any lower offspring survival may be due to pathogen transmission or potentially other factors

o In this experiment, it is difficult to distinguish among a few possibilities from the metrics shown in the figure: offspring death due to the pathogen, lower egg laying rates from females with higher pathogen loads, and lower egg lay totals from vials with fewer living females. However, there were not counts of the number of eggs laid per female in each vial. Ideally, there would be data on the number of eggs laid and viable offspring that ultimately developed per female from a given day. That would allow a truly direct comparison of offspring survival rates in each condition. Any additional data or information to clarify these points would be very helpful, otherwise there are significant caveats that should be addressed. It might help to include a table of the data to see how many vials of each category there were within each condition (viable, non-viable, excluded from analysis, etc.) or other underlying data. Also, if a full count of pupae was recorded for each vial, that data could be shown as a proxy for fitness impacts of the infection, although it still would not fully demonstrate survival rate differences without an egg number count

• Fig 4: why was this done as a male infection that is indirectly transferred to females, when females alone were tested in earlier experiments? With the possibility that results could be different between sexes, this could impact results

• Gut load measurements: Is it known for sure that none of the bacteria has gone to other tissues or become systemic? From the methods, it seems that the bacteria were measured from whole flies, not just guts

• Figure S3 comes before S2 in text, recommend swapping to avoid confusion

• Table S3/S4: confusing to use nipagin when it is mp in the rest of the text

• Line 372: should be "six-day-old individuals"

• For oral infection experiments (particularly figure 2), what were the ages and mating statuses of the flies?

Response to authors:

In this study, Henry et al. ask how the presence of methylparaben (mp) impacts the pathogenicity of oral infections with the Gram-negative bacteria *Pseudomonas entomophophila* (Pe) in *Drosophila melanogaster*. Generally, in the field, researchers deliver a significantly high OD (20-200 OD) of Pe to observe infection phenotypes. The authors provide a simple literature review showing that many studies using high inocula of Pe also fed their flies on diets with food preservatives like mp, while the few studies that use a low OD of Pe (less than OD 1) do not use any preservatives. Thus, the authors explore how five different inoculation doses of Pe (uninfected sham, OD 0.0001, OD 0.01, OD 1, and OD 50) impact infection survival of flies fed on diets with or without mp. Interestingly, diets without the preservative increase susceptibility to Pe infection even at lower doses than what is commonly used in the field. This increase in susceptibility to Pe infection in the absence of preservatives is ameliorated when flies are transferred daily to fresh food. Additionally, the authors show some evidence that transmission of Pe is increased when flies are fed preservative free food. This is an important study that interrogates how artifacts of typical fly lab media alter the pathogenicity of Pe. Some additional information would further improve the study.

Major comments

1. Figure 1: The authors show *in vitro* that 0.2% mp can completely inhibit growth after a 24-hour incubation, which suggests a bactericidal effect of the preservative. Line 121 in the text the authors state a personal observation of a potential bacteriostatic effect of only 6-hour exposure to the preservative. These methods aren't explained in the methods section nor is the motivation for this 6-hour timepoint explained. It is challenging to interpret a complete bactericidal effect based on the single *in vitro* experiment performed. Have the authors tried inoculating fresh cultures with the overnight cultures grown with methylparaben to see if anything can grow? Including a minimum inhibitory concentration (MIC) assay and a minimum bactericidal concentration (MBC) is critical for definitively supporting the authors claim that mp is bactericidal to Pe. Having a bactericidal and bacteriostatic antibiotic would be helpful positive controls if accessible.
2. Figure 2 and Figure S2: Aligned with the idea of whether mp is bactericidal or bacteriostatic, have the authors tried scoring the CFU counts in the food to see if they similarly match the CFU reflected in the fly? Or whether Pe is detectable in mp food? Understanding what the loads of Pe are in the food provides more certainty of whether mp is bactericidal and provides more context when interpreting the mother-larvae transmission experiments (e.g. the number of cells that larvae maybe exposed to).
3. Figure 3: How do the authors disentangle pathogen transmission from viability of the progeny? Hudson, AL, et al (2023, JEB) show that diets containing ~14%

protein (reflecting the protein content in the diet used in this study) lead to less viable progeny post oral infection with *P. aeruginosa* than progeny from flies fed on a high protein diet (~30% protein). They also show that the viability of progeny from infected females decrease over the time of the infection. Potentially, the lack of protein coupled with the stressors that come from carrying a higher bacteria burden in females on preservative free food could impair the viability of their progeny. An additional experiment such as having the eggs from uninfected females laid on vials from infected flies would be another form of evidence to decouple transmission of the pathogen from protein availability for the progeny/mother. Knowing how much CFUs are present in the food coupled with exposing progeny from uninfected mothers to infected vials could help clarify transmission to progeny.

4. Figure 4: I think the evidence that transmission of Pe is higher on mp absent food is weaker here since the authors only show how transmission is possible on the mp absent food. The authors cite Jothy et al (2018) for their oral infection procedure, and they also describe a method for measuring transmission by quantifying the amount of bacterial shedding. Have the authors tried using this method to assess how presence of mp impacts how much Pe infected flies shed?

Minor comments:

1. Figure 1S: Propionic acid has a similar result to the presence of mp where there are no detectable CFUs. What are the justifications for focusing on mp for the study when propionic acid shows to have a similar effect? It could be worthy to repeat some of the survival/CFU load measurements with the OD 1 and OD50 inoculum. This could be another study, so, alternatively, expanding discussion on lines 273-276 on the impact of propionic acid on Pe inoculum to include a need for future experiments.
2. Figure S2B shows that there is a low stable CFU of Pe in flies fed mp when transferred to fresh food, however, Figure 2B shows that there are no detectable loads when flies are not transferred to fresh vials. What is the authors interpretation of mp being bactericidal, however it does not clear the infection when flies are transferred to fresh food vials?
3. Figures 1 & 3: The methods section describes the statistical analyses used for both of these figures, however, the stats for these experiments are not present in the figure or the figure legend for both figures.
4. Line 127-128: For clarity for the reader, stating the dose range will be helpful when stating what their expectations for infection survival are. Can clarify here that "the expectations" are based on observations in the literature which authors

have already cited (lines 66-68), which should be cited here as well when referring to what one expects.

5. Line 328: The methods say that 20% methylparaben in 1% ethanol is added to a suspension of Pe in LB. But it does not state that the final concentration of the mixture is 0.2%, which matches the concentration of the fly food. What were the actual volumes used to calculate the final volume of this mixture then 0.2%? Based on the concentration provided, I believe you'd need to perform a 1:100-fold dilution, for example add 10 uL of 20% methylparaben to 990 uL of PE LB culture to achieve a 0.2% final concentration.
6. For the different OD doses, what are the approximate CFUs delivered per fly?
7. For all figure legends, should state what the concentration of mp is in the food.
8. Line 138 (and throughout text): For clarity to the readers, alternatively say "end of the infection time course" or "last day of sampling"?
9. Methods Adult-offspring pathogen transmission: how do the authors determine the larvae are dead (e.g. black/melanized, immobile)? Are these observations only of dead L3 larvae?
10. Line 317: Spell out the entire name of the species
11. Line 364: "Six day old" instead of "Six days old"
12. Line 405: What is the justification for using Bayesian framework for analyzing these datasets?

Responses to editor.

- Discussion lines 217-218: I think it's a stretch for the authors to state that only bacterial load drives Pe pathogenicity and that no Pe activity impacts survival phenotypes without any examples share of known mechanisms of Pe activity that cause host death/disease. Figure 2 shows that the -mp flies lead to similarly high pathogen loads that correspond to the death of the population. For example, OD1 and OD50 have similar Pe loads at the time the population crashes. Other infections inoculated at high doses like ecc15 do not cause the entire population to die like Pe does at similar loads, which suggests activity of the bacteria leading to high levels of acute lethality.
- Lines 255-260: I'm not entirely convinced that the paper cited being an example of mp exposure entirely explaining the route of infection differences. The

hemocoel and the gut are still two vastly different environments for Pe to be exposed to in the fly even if mp is not present in the hemolymph. For the claims the authors are making, it will be important for them to clarify their interpretations of what they mean by a “stronger immune responses against Pe” in context of the experiment they are referring to. For example, they could share alternative ideas of how lower ROS production to avoid sepsis syndrome requires description of the role of ROS in the gut. Overall, I felt like part of the discussion section makes large interpretations of what maybe occurring in other papers without strong evidence that could suggest otherwise.

Dear Editor, below we address, point-by-point, the constructive comments by the reviewers. Our responses are in blue and start with “###”. We carefully considered all mentioned issues and corrected the manuscript accordingly.

In this submission, we provide a clean version of the manuscript, but we also show the manuscript with tracked changes since the last submission to appreciate the changes.

Editor comment:

You will see that both reviewers valued the contributions of the work, but both also had some concerns about the limitations of the study and, in particular, that the discussion was over interpreted. You will see this in the comments below, but it also came up in their confidential comments to me directly. If you choose to revise I encourage you to carefully consider their comments, and to make sure that the discussion addresses other interpretations and makes it clear when speculation is based on data or may be more like a of evidence you favor for future work. It may not be necessary to remove text reflecting the latter if the caveats and alternatives are properly recognized, but the key is that they should be addressed.

We agree that some limitations of the study were not highlighted enough in the previous version of the manuscript. We completely reworked the discussion, in some cases to better acknowledge the limitations, or in other cases to remove parts with overinterpretation/speculation (see answers to reviewers' comments for details).

Reviewer #1 (Comments for the Author):

In this study, Henry et al. ask how the presence of methylparaben (mp) impacts the pathogenicity of oral infections with the Gram-negative bacteria *Pseudomonas entomophila* (Pe) in *Drosophila melanogaster*. Generally, in the field, researchers deliver a significantly high OD (20-200 OD) of Pe to observe infection phenotypes. The authors provide a simple literature review showing that many studies using high inocula of Pe also fed their flies on diets with food preservatives like mp, while the few studies that use a low OD of Pe (less than OD 1) do not use any preservatives. Thus, the authors explore how five different inoculation doses of Pe (uninfected sham, OD 0.0001, OD 0.01, OD 1, and OD 50) impact infection survival of flies fed on diets with or without mp. Interestingly, diets without the preservative increase susceptibility to Pe infection even at lower doses than what is commonly used in the field. This increase in susceptibility to Pe infection in the absence of preservatives is ameliorated when flies are transferred daily to fresh food. Additionally, the authors show some evidence that transmission of Pe is increased when flies are fed preservative free food. This is an important study that interrogates how artifacts of typical fly lab media alter the pathogenicity of Pe. Some additional information would further improve the study.

Major comments

1. Figure 1: The authors show in vitro that 0.2% mp can completely inhibit growth after a 24-hour incubation, which suggests a bactericidal effect of the preservative. Line 121 in the text the authors state a personal observation of a potential bacteriostatic effect of only 6-hour exposure to the preservative. These methods aren't explained in the methods section nor is the motivation for this 6-hour timepoint explained. It is challenging to interpret a complete bactericidal effect based on the single in vitro experiment performed. Have the authors tried inoculating fresh cultures with the overnight cultures grown with methylparaben to see if anything can grow? Including a minimum inhibitory concentration (MIC) assay and a minimum bactericidal concentration (MBC) is critical for definitively supporting the authors

claim that mp is bactericidal to Pe. Having a bactericidal and bacteriostatic antibiotic would be helpful positive controls if accessible.

We agree and thus we performed an additional experiment to estimate the MIC and the MBC of methylparaben, now included in Figure 1; the Methods and the Results parts have been modified accordingly. We removed the “personal observation” suggesting a bacteriostatic effect for short exposures to mp (6h), as we did not perform a replicated experiment to test that explicitly.

2. Figure 2 and Figure S2: Aligned with the idea of whether mp is bactericidal or bacteriostatic, have the authors tried scoring the CFU counts in the food to see if they similarly match the CFU reflected in the fly? Or whether Pe is detectable in mp food? Understanding what the loads of Pe are in the food provides more certainty of whether mp is bactericidal and provides more context when interpreting the mother-larvae transmission experiments (e.g. the number of cells that larvae maybe exposed to).

It would have been difficult to compare the load in the flies and in the food in a meaningful way. However, in response to this comment we have now performed a new experiment to explicitly test the presence of Pe in fly food after 20h being in contact with lowly infected adult flies. We indeed show that infected flies can contaminate their food with Pe, and that it can result in high “environmental” Pe loads, especially after few days of growth (see figure S3). This experiment also confirmed that, in the absence of mp, Pe can grow on fly food even in the absence of flies.

3. Figure 3: How do the authors disentangle pathogen transmission from viability of the progeny? Hudson, AL, et al (2023, JEB) show that diets containing ~14% protein (reflecting the protein content in the diet used in this study) lead to less viable progeny post oral infection with *P. aeruginosa* than progeny from flies fed on a high protein diet (~30% protein). They also show that the viability of progeny from infected females decrease over the time of the infection. Potentially, the lack of protein coupled with the stressors that come from carrying a higher bacteria burden in females on preservative free food could impair the viability of their progeny. An additional experiment such as having the eggs from uninfected females laid on vials from infected flies would be another form of evidence to decouple transmission of the pathogen from protein availability for the progeny/mother. Knowing how much CFUs are present in the food coupled with exposing progeny from uninfected mothers to infected vials could help clarify transmission to progeny.

This is a relevant remark. We performed the suggested experiment, allowing infected males to contaminate the food before manually transferring to this food eggs laid by uninfected females. This resulted in complete larval mortality in the absence of mp (but normal survival in its presence), consistent with indirect transmission of the pathogen from adults to offspring. These results have been included in the ms (supplementary figure S5). The new experiment mentioned in the response to the preceding comment also demonstrates the presence and growth of Pe on fly food.

4. Figure 4: I think the evidence that transmission of Pe is higher on mp absent food is weaker here since the authors only show how transmission is possible on the mp absent food. The authors cite Jothy et al (2018) for their oral infection procedure, and they also describe a method for measuring transmission by quantifying the amount of bacterial shedding. Have the authors tried using this method to assess how presence of mp impacts how much Pe infected flies shed?

We did not quantify shedding of bacterial by flies in different conditions; while it would provide an interesting detail, we feel it would go beyond the scope of this paper. The main point of figure 4 is to show that, in the absence of mp, a single adult initially infected with a low pathogen dose (OD =

0.01) can indirectly infect other healthy adults. Thus, Pe can potentially spread in a fly population, something that has not been previously reported, making it potentially an ecologically relevant pathogen. Indeed, we did not demonstrate that such a transmission does not happen (or happens less) on a food medium containing mp (although the new experiment in Figure S5 shows this for adult male to larvae transmission). We think that such a transmission resulting in mortality would be highly unlikely because:

- as the new experiment reported in figure 1 demonstrates, the concentration of mp in the fly food is lethal for Pe;
- in the presence of mp direct inoculation of the food with a 100x higher dose (OD = 1) did not generate any mortality.

It is possible that a male transferred to a new food vial would shed less Pe if the new food contained mp, if only because Pe in its gut would start to be negatively affected by the mp. We now acknowledge in the discussion the potential contribution of reduced bacterial shedding to inhibition of lateral transmission of Pe.

Minor comments:

1. Figure 1S: Propionic acid has a similar result to the presence of mp where there are no detectable CFUs. What are the justifications for focusing on mp for the study when propionic acid shows to have a similar effect? It could be worthy to repeat some of the survival/CFU load measurements with the OD 1 and OD50 inoculum. This could be another study, so, alternatively, expanding discussion on lines 273-276 on the impact of propionic acid on Pe inoculum to include a need for future experiments.

Propionic acid is not as frequently used as methylparaben in fly food medium recipes, which is the main reason why we focused on mp. We also tested the effect of propionic acid in vitro to broaden the scope of our findings and encourage cautious use of food preservatives in general, when working on insect-bacteria interactions. We have modified the text in the discussion to point out that the effects of other preservatives (including propionic acid) could be worth exploring.

2. Figure S2B shows that there is a low stable CFU of Pe in flies fed mp when transferred to fresh food, however, Figure 2B shows that there are no detectable loads when flies are not transferred to fresh vials. What is the authors interpretation of mp being bactericidal, however it does not clear the infection when flies are transferred to fresh food vials?

The figure the reviewer refers to is now 3B. Our detection threshold was 40 CFU per fly, and the average final Pe loads of mp+ flies, whether transferred daily or not, are both below that detection threshold (except for infection with OD = 50). This means that in most replicates no colonies were observed at all, with a few replicates with 1-2 Pe colonies growing in the least diluted plating. This implies that indeed a few Pe cells were still alive in +mp conditions. We can only speculate that the fly bottle + food + flies generate a more complex and heterogeneous environment for Pe than that the in vitro conditions, with some refugia where the bacteria are somewhat protected from the full effects of mp and can persist in small numbers (e.g., on bottle wall or the surface of the foam plug). We now discuss this residual Pe presence on +mp diet, and provide this potential explanation in the manuscript.

3. Figures 1 & 3: The methods section describes the statistical analyses used for both of these figures, however, the stats for these experiments are not present in the figure or the figure legend for both figures.

We now have added an asterisk and the associated reference in the caption, to indicate non-overlapping credible intervals in figure 1 (now figure S1) as well as in new figures, when required. For

figure 3 (now figure 4), we do not perform pairwise comparisons, and we only tested the general effect of mp in interaction with the Pe dose, on the viability of the generation F1. For this reason, we only include the statistical results in the text for figure 3.

4. Line 127-128: For clarity for the reader, stating the dose range will be helpful when stating what their expectations for infection survival are. Can clarify here that "the expectations" are based on observations in the literature which authors have already cited (lines 66-68), which should be cited here as well when referring to what one expects.

This is a good suggestion; we modified the text accordingly.

5. Discussion lines 217-218: I think it's a stretch for the authors to state that only bacterial load drives Pe pathogenicity and that no Pe activity impacts survival phenotypes without any examples share of known mechanisms of Pe activity that cause host death/disease. Figure 2 shows that the -mp flies lead to similarly high pathogen loads that correspond to the death of the population. For example, OD1 and OD50 have similar Pe loads at the time the population crashes. Other infections inoculated at high doses like ecc15 do not cause the entire population to die like Pe does at similar loads, which suggests activity of the bacteria leading to high levels of acute lethality.

It was not our intention to imply that only the bacterial load determined host mortality. It is obviously very likely to depend on whether the Pe are metabolically active, and e.g. whether or not they secrete the proteases that are one of the virulence factors.

We reworked the ambiguous text in the discussion. We now simply say that our results suggest mortality can be induced by weakened or dying Pe cells, as long as the load is sufficiently high (backed with the reference to Paulo et al., 2024). The new text is as follows:

“As has been extensively studied, Pe kills the host by compromising gut homeostasis and integrity through action of bacterial toxins and inhibition of epithelial renewal, possibly exacerbated by ROS produced by the gut as a part of the immune response (Bou Sleiman et al., 2015; Chakrabarti et al., 2012; Opota et al., 2011; Vijendravarma et al., 2015). It is possible that at these very high doses, even weakened or dying Pe cells induce sufficient harm either directly (e.g. by releasing toxins) or via inducing harmful immune response. Consistent with this view, high mortality in flies can be induced by repeated ingestion of large amounts of heat-killed Pe (Paulo et al., 2024).”

6. Lines 255-260: I'm not entirely convinced that the paper cited being an example of mp exposure entirely explaining the route of infection differences. The hemocoel and the gut are still two vastly different environments for Pe to be exposed to in the fly even if mp is not present in the hemolymph. For these claims, the authors should clarify their interpretations of what they mean by a "stronger immune responses against Pe" in context of the experiment they are referring to. For example, the authors could share alternative ideas of how lower ROS production to avoid sepsis syndrome requires description of the role of ROS in the gut. Overall, I felt like part of the discussion section, I makes large interpretations of what maybe occurring in other papers without strong evidence that could suggest otherwise.

We agree that the discussion comparing intestinal with systemic infection was too speculative; we removed it completely.

7. Line 328: The methods say that 20% methylparaben in 1% ethanol is added to a

suspension of Pe in LB. But it does not state that the final concentration of the mixture is 0.2%, which matches the concentration of the fly food. What were the actual volumes used to calculate the final volume of this mixture then 0.2%? Based on the concentration provided, I believe you'd need to perform a 1:100-fold dilution, for example add 10 uL of 20% methylparaben to 990 uL of PE LB culture to achieve a 0.2% final concentration.

This part has changed a lot, part of it has been moved to the supplementary material, and we now mention in the Methods and in the figure caption the final concentration of mp (0.2%). We don't think that including the exact volumes used in the experiment is relevant. After careful re-reading of the methods, we do not find any calculation errors.

8. For the different OD doses, what are the approximate CFUs delivered per fly?

We can only estimate that as it was not directly measured. A rough estimation is that flies eat approx. 1uL of food per day. They were exposed to the infection for one day, so with $OD1 = 1.10^9$ CFU/mL, it means each fly ingested 1.10^6 cells of Pe in this condition. In the lowest OD (OD 0.0001), they ingested about 10 cells of Pe. These estimations are higher than the measured CFU in flies after 1d of infection shown in figure 2, but CFU in flies are obviously reduced by the immune system and by their excretion with feces. Because this is a rough estimate whose assumptions can be questioned, we prefer not to include it in the manuscript.

9. For all figure legends, should state what the concentration of mp is in the food.

We now mention the mp concentration in the +mp food in figure legends.

10. Line 138 (and throughout text): For clarity to the readers, alternatively say "end of the infection time course" or "last day of sampling"?

We made the requested change (we used "on the last day of sampling" to replace most instances of "at final time").

11. Methods Adult-offspring pathogen transmission: how do the authors determine the larvae are dead (e.g. black/melanized, immobile)? Are these observations only of dead L3 larvae?

Dead larvae were only L1/L2 and were immobile and melanized. But we assessed mortality by looking at the presence/absence of pupae. The newly added illustrative photos in figure 4A help the reader to see the effects of Pe on the offspring on -mp diet.

12. Line 317: Spell out the entire name of the species

We made the requested change.

13. Line 364: "Six day old" instead of "Six days old"

We made the requested change (adding hyphens).

14. Line 405: What is the justification for using Bayesian framework for analyzing these datasets?

Fundamentally, there should not be any justification required to pick a Bayesian approach instead of frequentist as both are completely valid here. Using Bayesian framework is also a way to slowly move away from the cult of p-values and all the issues associated. That being said, we also opted for Bayesian methods because these approaches are generally performing better with the kind of data we had (e.g. the large numbers and zeros for CFU, or the excess of zeros/ones in the offspring

transmission generating infinite parameter estimates in the models). For CFUs specifically, we also had trouble meeting the model requirements (in terms of residual distribution) with the frequentist models.

Reviewer #2 (Comments for the Author):

In this manuscript, Henry et al. demonstrate that the lethality of a *Drosophila* pathogen, *Pseudomonas entomophila*, is strongly affected by the presence or absence of preservatives that are nearly universally used in fly media. In the field of *Drosophila* immunity, *P. entomophila* is a very commonly used pathogen in research, but it is often used at very high concentrations in order to kill flies when infected orally. Here, the authors show that methylparaben (and propionic acid), an antimicrobial preservative that is used in most standard fly food recipes, acts to reduce the among individuals. This study is important to the insect immunity research community lethality of the bacteria. In fly food media prepared without the preservative, the pathogen is lethal at much lower concentrations. The authors also perform experiments that indicate the preservative may reduce transmission of pathogens as these results indicate that ubiquitous ingredients in fly food may inadvertently affect the interpretation of some experimental results and that more ecologically relevant experiments and results may be possible in the lab (ie, with pathogen concentrations at a more natural level) without the preservatives. Overall, the manuscript is straightforward, with some additional input needed on the points below:

- For all of the experiments, the level of replication was somewhat unclear. The number and types of samples within each treatment group and data point were laid out, but it was unclear if full experiments with all treatment conditions were repeated at different times. Were experiments repeated on different days with new biological samples? If experiments were repeated, it would help to state so in methods/captions. If not, it would be important to address or discuss this

We agree that it was not always easy to follow the replication level throughout the manuscript. We clarified the replication level everywhere we found it necessary.

There were three (now five after revisions) separate experiments in total reported in this manuscript, and all replicates of a given experiment were performed at the same time. However, the different experiments were performed at different times, using independent *Pe* inocula, and their results are generally consistent with one another with respect to the major conclusions of the study. Furthermore, before the experiments reported here, we performed pilot experiments that first alerted us to the role of mp in protecting flies from *Pe*. Thus, we have no reason to expect a qualitatively different outcome if we repeat the experiments under exactly the same conditions. However, we do agree that changing the conditions (e.g., the microbiota as in Barron et al., 2024) or using a different strain of *Pe* could lead to different results (our strain is, however, the one most frequently used *Drosophila* research). We added an acknowledgement of this in the Discussion.

- Fig 3 & parent to offspring pathogen transmission experiment:

- o The scoring of offspring viability was based on the number of visible dead larval and pupal offspring in each vial six days after adults laid eggs. It is difficult to understand how accurately this could be done as standard fly media is an opaque brown color which would obscure many or most larvae unless they happen to be at the edge of the vial. How were larvae viewed reliably and not missed in any vials and how was it determined if the pupae had died by day 6, when most would still be early in pupal development? Also, why was the value of 5 chosen as the cutoff for viability?

In this “pathogen transmission” part of the experiment, we took an opportunity offered by the eggs laid by females in the experiment whose main purpose was to study the effect of daily fly transfers on Pe-induced mortality. While we did not know the initial number of eggs and thus could not quantify survival, the outcome was (mostly) binary: either the offspring (larvae/pupae) in a particular vial were perfectly healthy, with many alive individuals (generally >50), or the vials mostly contained many visible dead melanized larvae on the vial sides and a maximum of 1 or 2 lucky survivors pupating. Because the dichotomy was extremely clear, we decided to classify the vials as “viable” (full of pupae) or “non-viable” (dead larvae). See the new figure 4A for illustration. We anticipated a possible criticism of this procedure: one could think that Pe-infected females are not laying eggs anymore due to the infection, or that no eggs would be laid in vials with too many dead females. Because of that, we introduced this arbitrary minimum threshold of 5 visible pupae/dead larvae, to avoid counting vials that never contained eggs.

The reviewer also mentions possible mortality at the pupal stage. It is true that we cannot exclude late mortality due to the pathogen with our experimental procedure. Yet, a vast majority of the offspring mortality was happening at the early larval stages (L1, L2), rarely in L3, and all individuals reaching the pupal stage emerged as viable adults. Because we did not record data about this (we only have personal observations), we changed all instances of “offspring viability” into “pupation success” or “vial viability”, depending on the context.

We tried to improve the description of our methods for the assessment of offspring viability and we think the new experiment reported in figure S5 resolves most of the methodological concerns expressed by the reviewer.

o Line 176-177: The text describes this experiment as measuring egg to pupal survival rate, but it seems that eggs were not counted and the metric used is not really egg to pupal survival, but rather number of visible dead offspring.

We clarified the vocabulary, and we now mention “vial viability” as the metric (see comment above).

o The results show lower offspring viability without the preservative and the authors suggest that this indicates transmission of the pathogen from adult to offspring. However, there is no direct confirmation that the pathogen was passed on to the next generation since offspring were not tested for the pathogen (measuring CFU's etc.), and were only counted for visible survival. The language should be amended to reflect that any lower offspring survival may be due to pathogen transmission or potentially other factors

This comment is in line with the comments from reviewer 1. We performed two new experiments, showing that 1) adult flies initially infected with a low Pe dose could contaminate the food with Pe, and 2) that eggs transferred to a food where infected individuals stayed for 24h showed great mortality only in mp- conditions. (figures S3 and S4)

o In this experiment, it is difficult to distinguish among a few possibilities from the metrics shown in the figure: offspring death due to the pathogen, lower egg laying rates from females with higher pathogen loads, and lower egg lay totals from vials with fewer living females. However, there were not counts of the number of eggs laid per female in each vial. Ideally, there would be data on the number of eggs laid and viable offspring that ultimately developed per female from a given day. That would allow a truly direct comparison of offspring survival rates in each condition. Any additional data or information to clarify these points would be very helpful, otherwise there are significant caveats that should be addressed. It might help

to include a table of the data to see how many vials of each category there were within each condition (viable, non-viable, excluded from analysis, etc.) or other underlying data. Also, if a full count of pupae was recorded for each vial, that data could be shown as a proxy for fitness impacts of the infection, although it still would not fully demonstrate survival rate differences without an egg number count

We performed an additional experiment to answer this issue, see comment above and replies to reviewer 1.

- Fig 4: why was this done as a male infection that is indirectly transferred to females, when females alone were tested in earlier experiments? With the possibility that results could be different between sexes, this could impact results

We now acknowledge a possible effect of sex in the discussion. In our new experiment (figure S3), we tested medium contamination resulting from male or female Pe-infected individuals. Although our statistical power was too limited to properly test the effect of sex on transmission, we could not observe an overwhelmingly different pattern depending on sex. Anyways, using males for the indirect infections in our experiments was mainly for technical/practical reasons:

- 1) Because we demonstrated that the transmission of Pe to the offspring was entirely killing the next generation (see fig 3), having females for the infections means that infected vs sham vials would be different in two ways (having Pe, vs no Pe, and having no larvae vs no larvae). This confounding factor was delayed by one day, using males for infections.
- 2) Mated females lay eggs, and the crawling larvae tend to make mortality measurements hard, as the vials get dirty.
- 3) Larvae tend to eat weakened adult flies

Ideally, we would have used virgin females, but one could still argue a possible effect of reproduction (see Gupta et al 2013, cited in the ms).

- Gut load measurements: Is it known for sure that none of the bacteria has gone to other tissues or become systemic? From the methods, it seems that the bacteria were measured from whole flies, not just guts

It has been shown in previous studies (e.g. Vijendravarma, R. K., Narasimha, S., Chakrabarti, S., Babin, A., Kolly, S., Lemaitre, B. and Kawecki, T. J. (2015). Gut physiology mediates a trade-off between adaptation to malnutrition and susceptibility to food-borne pathogens. *Ecology Letters* 18, 1078–1086.) that the loss of the gut integrity was a sign of imminent death of the individuals. Therefore, it is likely that the alive individuals we sampled still had functional gut barrier and were only infected in the gut, but we cannot be absolutely certain of that. We now acknowledge this limitation in the methods.

- Figure S3 comes before S2 in text, recommend swapping to avoid confusion

Correct. We made the requested change.

- Table S3/S4: confusing to use nipagin when it is mp in the rest of the text

We changed all instances of “nipagin” into “mp” in suppl tables.

- Line 372: should be "six-day-old individuals"

We made the requested change.

- For oral infection experiments (particularly figure 2), what were the ages and mating statuses of the flies?

This information is already indicated in the methods. Flies were 6 day-old mated females at time 0 in figure 2.

Re: Spectrum03065-24R1 (Standard ingredient of *Drosophila* medium reduces transmission and virulence of the gut pathogen *Pseudomonas entomophila*)

Dear Dr. Youn Henry:

Thank you for the privilege of reviewing your work. I appreciate the time you took in revision, and I look forward to passing it back to the reviewers to get their comments. Before I send it out for review, could you please resubmit all of the same files as before, except please add line numbers from the 'clean' version of the manuscript to each response? That way the reviewers will know where to find the change you indicate. It will make their job easier if you can do that before I send out for review.

Revision Guidelines

Sincerely,
John Chaston
Editor
Microbiology Spectrum

Dear Editor, below we address, point-by-point, the constructive comments by the reviewers. Our responses are in blue and start with “###”. We carefully considered all mentioned issues and corrected the manuscript accordingly.

In this submission, we provide a clean version of the manuscript, but we also show the manuscript with tracked changes since the last submission to appreciate the changes.

Editor comment:

You will see that both reviewers valued the contributions of the work, but both also had some concerns about the limitations of the study and, in particular, that the discussion was over interpreted. You will see this in the comments below, but it also came up in their confidential comments to me directly. If you choose to revise I encourage you to carefully consider their comments, and to make sure that the discussion addresses other interpretations and makes it clear when speculation is based on data or may be more like a of evidence you favor for future work. It may not be necessary to remove text reflecting the latter if the caveats and alternatives are properly recognized, but the key is that they should be addressed.

We agree that some limitations of the study were not highlighted enough in the previous version of the manuscript. We completely reworked the discussion, in some cases to better acknowledge the limitations, or in other cases to remove parts with overinterpretation/speculation (see answers to reviewers' comments for details).

Reviewer #1 (Comments for the Author):

In this study, Henry et al. ask how the presence of methylparaben (mp) impacts the pathogenicity of oral infections with the Gram-negative bacteria *Pseudomonas entomophila* (Pe) in *Drosophila melanogaster*. Generally, in the field, researchers deliver a significantly high OD (20-200 OD) of Pe to observe infection phenotypes. The authors provide a simple literature review showing that many studies using high inocula of Pe also fed their flies on diets with food preservatives like mp, while the few studies that use a low OD of Pe (less than OD 1) do not use any preservatives. Thus, the authors explore how five different inoculation doses of Pe (uninfected sham, OD 0.0001, OD 0.01, OD 1, and OD 50) impact infection survival of flies fed on diets with or without mp. Interestingly, diets without the preservative increase susceptibility to Pe infection even at lower doses than what is commonly used in the field. This increase in susceptibility to Pe infection in the absence of preservatives is ameliorated when flies are transferred daily to fresh food. Additionally, the authors show some evidence that transmission of Pe is increased when flies are fed preservative free food. This is an important study that interrogates how artifacts of typical fly lab media alter the pathogenicity of Pe. Some additional information would further improve the study.

Major comments

1. Figure 1: The authors show in vitro that 0.2% mp can completely inhibit growth after a 24-hour incubation, which suggests a bactericidal effect of the preservative. Line 121 in the text the authors state a personal observation of a potential bacteriostatic effect of only 6-hour exposure to the preservative. These methods aren't explained in the methods section nor is the motivation for this 6-hour timepoint explained. It is challenging to interpret a complete bactericidal effect based on the single in vitro experiment performed. Have the authors tried inoculating fresh cultures with the overnight cultures grown with methylparaben to see if anything can grow? Including a minimum inhibitory concentration (MIC) assay and a minimum bactericidal concentration (MBC) is critical for definitively supporting the authors

claim that mp is bactericidal to Pe. Having a bactericidal and bacteriostatic antibiotic would be helpful positive controls if accessible.

We agree and thus we performed an additional experiment to estimate the MIC and the MBC of methylparaben, now included in Figure 1; the Methods (L338-344) and the Results (L111-123) parts have been modified accordingly. We removed the “personal observation” suggesting a bacteriostatic effect for short exposures to mp (6h), as we did not perform a replicated experiment to test that explicitly.

2. Figure 2 and Figure S2: Aligned with the idea of whether mp is bactericidal or bacteriostatic, have the authors tried scoring the CFU counts in the food to see if they similarly match the CFU reflected in the fly? Or whether Pe is detectable in mp food? Understanding what the loads of Pe are in the food provides more certainty of whether mp is bactericidal and provides more context when interpreting the mother-larvae transmission experiments (e.g. the number of cells that larvae maybe exposed to).

It would have been difficult to compare the load in the flies and in the food in a meaningful way. However, in response to this comment we have now performed a new experiment to explicitly test the presence of Pe in fly food after 20h being in contact with lowly infected adult flies (L163-168; L420-434). We indeed show that infected flies can contaminate their food with Pe, and that it can result in high “environmental” Pe loads, especially after few days of growth (see figure S3). This experiment also confirmed that, in the absence of mp, Pe can grow on fly food even in the absence of flies.

3. Figure 3: How do the authors disentangle pathogen transmission from viability of the progeny? Hudson, AL, et al (2023, JEB) show that diets containing ~14% protein (reflecting the protein content in the diet used in this study) lead to less viable progeny post oral infection with *P. aeruginosa* than progeny from flies fed on a high protein diet (~30% protein). They also show that the viability of progeny from infected females decrease over the time of the infection. Potentially, the lack of protein coupled with the stressors that come from carrying a higher bacteria burden in females on preservative free food could impair the viability of their progeny. An additional experiment such as having the eggs from uninfected females laid on vials from infected flies would be another form of evidence to decouple transmission of the pathogen from protein availability for the progeny/mother. Knowing how much CFUs are present in the food coupled with exposing progeny from uninfected mothers to infected vials could help clarify transmission to progeny.

This is a relevant remark. We performed the suggested experiment, allowing infected males to contaminate the food before manually transferring to this food eggs laid by uninfected females. This resulted in complete larval mortality in the absence of mp (but normal survival in its presence), consistent with indirect transmission of the pathogen from adults to offspring. These results have been included in the ms (L 182-190; supplementary figure S4). The new experiment mentioned in the response to the preceding comment also demonstrates the presence and growth of Pe on fly food.

4. Figure 4: I think the evidence that transmission of Pe is higher on mp absent food is weaker here since the authors only show how transmission is possible on the mp absent food. The authors cite Jothy et al (2018) for their oral infection procedure, and they also describe a method for measuring transmission by quantifying the amount of bacterial shedding. Have the authors tried using this method to assess how presence of mp impacts how much Pe infected flies shed?

We did not quantify shedding of bacterial by flies in different conditions; while it would provide an interesting detail, we feel it would go beyond the scope of this paper. The main point of figure 4

(now figure 5) is to show that, in the absence of mp, a single adult initially infected with a low pathogen dose (OD = 0.01) can indirectly infect other healthy adults. Thus, Pe can potentially spread in a fly population, something that has not been previously reported, making it potentially an ecologically relevant pathogen. Although we have not formally demonstrated that adult-adult transmission happens less on a food medium containing mp (the new experiment in Figure S5 only shows this for adult male to larvae transmission), we think that such a transmission resulting in mortality would be highly unlikely because:

- as the new experiment reported in figure 1 demonstrates, the concentration of mp in the fly food is lethal for Pe;
- in the presence of mp direct inoculation of the food with a 100x higher dose (OD = 1) did not generate any mortality.

It is possible that a male transferred to a new food vial would shed less Pe if the new food contained mp, if only because Pe in its gut would start to be negatively affected by the mp. We now acknowledge in the discussion the potential contribution of reduced bacterial shedding to inhibition of lateral transmission of Pe (L266-268).

Minor comments:

1. Figure 1S: Propionic acid has a similar result to the presence of mp where there are no detectable CFUs. What are the justifications for focusing on mp for the study when propionic acid shows to have a similar effect? It could be worthy to repeat some of the survival/CFU load measurements with the OD 1 and OD50 inoculum. This could be another study, so, alternatively, expanding discussion on lines 273-276 on the impact of propionic acid on Pe inoculum to include a need for future experiments.

Propionic acid is not as frequently used as methylparaben in fly food medium recipes, which is the main reason why we focused on mp. We also tested the effect of propionic acid in vitro to broaden the scope of our findings and encourage cautious use of food preservatives in general, when working on insect-bacteria interactions. We have modified the text in the discussion to point out that the effects of other preservatives (including propionic acid) could be worth exploring (L294-301).

2. Figure S2B shows that there is a low stable CFU of Pe in flies fed mp when transferred to fresh food, however, Figure 2B shows that there are no detectable loads when flies are not transferred to fresh vials. What is the authors interpretation of mp being bactericidal, however it does not clear the infection when flies are transferred to fresh food vials?

The figure the reviewer refers to is now 3B. Our detection threshold was 40 CFU per fly, and the average final Pe loads of mp+ flies, whether transferred daily or not, are both below that detection threshold (except for infection with OD = 50). This means that in most replicates no colonies were observed at all, with a few replicates with 1-2 Pe colonies growing in the least diluted plating. This implies that indeed a few Pe cells were still alive in +mp conditions. We can only speculate that the fly bottle + food + flies generate a more complex and heterogeneous environment for Pe than that the in vitro conditions, with some refugia where the bacteria are somewhat protected from the full effects of mp and can persist in small numbers (e.g., on bottle wall or the surface of the foam plug). We now discuss this residual Pe presence on +mp diet, and provide this potential explanation in the manuscript (L229-234).

3. Figures 1 & 3: The methods section describes the statistical analyses used for both of these figures, however, the stats for these experiments are not present in the figure or the figure legend for both figures.

We now have added an asterisk and the associated reference in the caption, to indicate non-overlapping credible intervals in figure 1 (now figure S1) as well as in new figures, when required. For figure 3 (now figure 4), we do not perform pairwise comparisons, and we only tested the general effect of mp in interaction with the Pe dose, on the viability of the generation F1. For this reason, we only include the statistical results in the text for current figure 4.

4. Line 127-128: For clarity for the reader, stating the dose range will be helpful when stating what their expectations for infection survival are. Can clarify here that "the expectations" are based on observations in the literature which authors have already cited (lines 66-68), which should be cited here as well when referring to what one expects.

This is a good suggestion; we modified the text accordingly (L125 & 127).

5. Discussion lines 217-218: I think it's a stretch for the authors to state that only bacterial load drives Pe pathogenicity and that no Pe activity impacts survival phenotypes without any examples share of known mechanisms of Pe activity that cause host death/disease. Figure 2 shows that the -mp flies lead to similarly high pathogen loads that correspond to the death of the population. For example, OD1 and OD50 have similar Pe loads at the time the population crashes. Other infections inoculated at high doses like ecc15 do not cause the entire population to die like Pe does at similar loads, which suggests activity of the bacteria leading to high levels of acute lethality.

It was not our intention to imply that only the bacterial load determined host mortality. It is obviously very likely to depend on whether the Pe are metabolically active, and e.g. whether or not they secrete the proteases that are one of the virulence factors.

We reworked the ambiguous text in the discussion. We now simply say that our results suggest mortality can be induced by weakened or dying Pe cells, as long as the load is sufficiently high (backed with the reference to Paulo et al., 2024). The new text is as follows (L213-220):

“As has been extensively studied, Pe kills the host by compromising gut homeostasis and integrity through action of bacterial toxins and inhibition of epithelial renewal, possibly exacerbated by ROS produced by the gut as a part of the immune response (Bou Sleiman et al., 2015; Chakrabarti et al., 2012; Opota et al., 2011; Vijendravarma et al., 2015). It is possible that at these very high doses, even weakened or dying Pe cells induce sufficient harm either directly (e.g. by releasing toxins) or via inducing harmful immune response. Consistent with this view, high mortality in flies can be induced by repeated ingestion of large amounts of heat-killed Pe (Paulo et al., 2024).”

6. Lines 255-260: I'm not entirely convinced that the paper cited being an example of mp exposure entirely explaining the route of infection differences. The hemocoel and the gut are still two vastly different environments for Pe to be exposed to in the fly even if mp is not present in the hemolymph. For these claims, the authors should clarify their interpretations of what they mean by a "stronger immune responses against Pe" in context of the experiment they are referring to. For example, the authors could share alternative ideas of how lower ROS production to avoid sepsis syndrome requires description of the role of ROS in the gut. Overall, I felt like part of the discussion section makes large interpretations of what maybe occurring in other papers without strong evidence that could suggest otherwise.

We agree that the discussion comparing intestinal with systemic infection was too speculative; we removed it completely.

7. Line 328: The methods say that 20% methylparaben in 1% ethanol is added to a suspension of Pe in LB. But it does not state that the final concentration of the mixture is 0.2%, which matches the concentration of the fly food. What were the actual volumes used to calculate the final volume of this mixture then 0.2%? Based on the concentration provided, I believe you'd need to perform a 1:100-fold dilution, for example add 10 uL of 20% methylparaben to 990 uL of PE LB culture to achieve a 0.2% final concentration.

This part has changed a lot, part of it has been moved to the supplementary material, and we now mention in the Methods (L340) and in the figure caption the final concentration of mp (0.2%). We don't think that including the exact volumes used in the experiment is relevant. After careful re-reading of the methods, we do not find any calculation errors.

8. For the different OD doses, what are the approximate CFUs delivered per fly?

We can only estimate that as it was not directly measured. A rough estimation is that flies eat approx. 1uL of food per day. They were exposed to the infection for one day, so with $OD1 = 1.10^9$ CFU/mL, it means each fly ingested 1.10^6 cells of Pe in this condition. In the lowest OD (OD 0.0001), they ingested about 10 cells of Pe. These estimations are higher than the measured CFU in flies after 1d of infection shown in figure 2, but CFU in flies are obviously reduced by the immune system and by their excretion with feces. Because this is a rough estimate whose assumptions can be questioned, we prefer not to include it in the manuscript.

9. For all figure legends, should state what the concentration of mp is in the food.

We now mention the mp concentration in the +mp food in figure legends (L664, 678, 681).

10. Line 138 (and throughout text): For clarity to the readers, alternatively say "end of the infection time course" or "last day of sampling"?

We made the requested change (we used "on the last day of sampling" to replace most instances of "at final time"). (L130, 137, 142)

11. Methods Adult-offspring pathogen transmission: how do the authors determine the larvae are dead (e.g. black/melanized, immobile)? Are these observations only of dead L3 larvae?

Dead larvae were only L1/L2 and were immobile and melanized. But we assessed mortality by looking at the presence/absence of pupae. The newly added illustrative photos in figure 4A help the reader to see the effects of Pe on the offspring on -mp diet.

12. Line 317: Spell out the entire name of the species

We made the requested change (L329).

13. Line 364: "Six day old" instead of "Six days old"

We made the requested change (adding hyphens). (L374)

14. Line 405: What is the justification for using Bayesian framework for analyzing these datasets?

Fundamentally, there should not be any justification required to pick a Bayesian approach instead of frequentist as both are completely valid here. Using Bayesian framework is also a way to slowly move away from the cult of p-values and all the issues associated. That being said, we also opted for

Bayesian methods because these approaches are generally performing better with the kind of data we had (e.g. the large numbers and zeros for CFU, or the excess of zeros/ones in the offspring transmission generating infinite parameter estimates in the models). For CFUs specifically, we also had trouble meeting the model requirements (in terms of residual distribution) with the frequentist models.

Reviewer #2 (Comments for the Author):

In this manuscript, Henry et al. demonstrate that the lethality of a *Drosophila* pathogen, *Pseudomonas entomophila*, is strongly affected by the presence or absence of preservatives that are nearly universally used in fly media. In the field of *Drosophila* immunity, *P. entomophila* is a very commonly used pathogen in research, but it is often used at very high concentrations in order to kill flies when infected orally. Here, the authors show that methylparaben (and propionic acid), an antimicrobial preservative that is used in most standard fly food recipes, acts to reduce the among individuals. This study is important to the insect immunity research community lethality of the bacteria. In fly food media prepared without the preservative, the pathogen is lethal at much lower concentrations. The authors also perform experiments that indicate the preservative may reduce transmission of pathogens as these results indicate that ubiquitous ingredients in fly food may inadvertently affect the interpretation of some experimental results and that more ecologically relevant experiments and results may be possible in the lab (ie, with pathogen concentrations at a more natural level) without the preservatives. Overall, the manuscript is straightforward, with some additional input needed on the points below:

- For all of the experiments, the level of replication was somewhat unclear. The number and types of samples within each treatment group and data point were laid out, but it was unclear if full experiments with all treatment conditions were repeated at different times. Were experiments repeated on different days with new biological samples? If experiments were repeated, it would help to state so in methods/captions. If not, it would be important to address or discuss this

We agree that it was not always easy to follow the replication level throughout the manuscript. We clarified the replication level everywhere we found it necessary (L657, 667, 688, 698).

There were three (now five after revisions) separate experiments in total reported in this manuscript, and all replicates of a given experiment were performed at the same time. However, the different experiments were performed at different times, using independent *Pe* inocula, and their results are generally consistent with one another with respect to the major conclusions of the study. Furthermore, before the experiments reported here, we performed pilot experiments that first alerted us to the role of mp in protecting flies from *Pe*. Thus, we have no reason to expect a qualitatively different outcome if we repeat the experiments under exactly the same conditions. However, we do agree that changing the conditions (e.g., the microbiota as in Barron et al., 2024) or using a different strain of *Pe* could lead to different results (our strain is, however, the one most frequently used *Drosophila* research). We added an acknowledgement of this in the Discussion (L284-287).

- Fig 3 & parent to offspring pathogen transmission experiment:

- o The scoring of offspring viability was based on the number of visible dead larval and pupal offspring in each vial six days after adults laid eggs. It is difficult to understand how accurately this could be done as standard fly media is an opaque brown color which would obscure many or most larvae unless they happen to be at the edge of the vial. How were larvae viewed reliably and not missed in any vials and how was it determined if the pupae

had died by day 6, when most would still be early in pupal development? Also, why was the value of 5 chosen as the cutoff for viability?

In this “pathogen transmission” part of the experiment, we took an opportunity offered by the eggs laid by females in the experiment whose main purpose was to study the effect of daily fly transfers on Pe-induced mortality. While we did not know the initial number of eggs and thus could not quantify survival, the outcome was (mostly) binary: either the offspring (larvae/pupae) in a particular vial were perfectly healthy, with many alive individuals (generally >50), or the vials mostly contained many visible dead melanized larvae on the vial sides and a maximum of 1 or 2 lucky survivors pupating. Because the dichotomy was extremely clear, we decided to classify the vials as “viable” (full of pupae) or “non-viable” (dead larvae). See the new figure 4A for illustration. We anticipated a possible criticism of this procedure: one could think that Pe-infected females are not laying eggs anymore due to the infection, or that no eggs would be laid in vials with too many dead females. Because of that, we introduced this arbitrary minimum threshold of 5 visible pupae/dead larvae, to avoid counting vials that never contained eggs.

The reviewer also mentions possible mortality at the pupal stage. It is true that we cannot exclude late mortality due to the pathogen with our experimental procedure. Yet, a vast majority of the offspring mortality was happening at the early larval stages (L1, L2), rarely in L3, and all individuals reaching the pupal stage emerged as viable adults. Because we did not record data about this (we only have personal observations), we changed all instances of “offspring viability” into “pupation success” or “vial viability”, depending on the context.

We tried to improve the description of our methods for the assessment of offspring viability (L392-401) and we think the new experiment reported in figure S5 resolves most of the methodological concerns expressed by the reviewer.

o Line 176-177: The text describes this experiment as measuring egg to pupal survival rate, but it seems that eggs were not counted and the metric used is not really egg to pupal survival, but rather number of visible dead offspring.

We clarified the vocabulary, and we now mention “vial viability” as the metric (see comment above). (L177-181)

o The results show lower offspring viability without the preservative and the authors suggest that this indicates transmission of the pathogen from adult to offspring. However, there is no direct confirmation that the pathogen was passed on to the next generation since offspring were not tested for the pathogen (measuring CFU's etc.), and were only counted for visible survival. The language should be amended to reflect that any lower offspring survival may be due to pathogen transmission or potentially other factors

This comment is in line with the comments from reviewer 1. We performed two new experiments, showing that 1) adult flies initially infected with a low Pe dose could contaminate the food with Pe, and 2) that eggs transferred to a food where infected individuals stayed for 24h showed great mortality only in mp- conditions. (figures S3 and S4)

o In this experiment, it is difficult to distinguish among a few possibilities from the metrics shown in the figure: offspring death due to the pathogen, lower egg laying rates from females with higher pathogen loads, and lower egg lay totals from vials with fewer living females. However, there were not counts of the number of eggs laid per female in each vial. Ideally, there would be data on the number of eggs laid and viable offspring that ultimately developed

per female from a given day. That would allow a truly direct comparison of offspring survival rates in each condition. Any additional data or information to clarify these points would be very helpful, otherwise there are significant caveats that should be addressed. It might help to include a table of the data to see how many vials of each category there were within each condition (viable, non-viable, excluded from analysis, etc.) or other underlying data. Also, if a full count of pupae was recorded for each vial, that data could be shown as a proxy for fitness impacts of the infection, although it still would not fully demonstrate survival rate differences without an egg number count

We performed an additional experiment to answer this issue, see comment above and replies to reviewer 1.

- Fig 4: why was this done as a male infection that is indirectly transferred to females, when females alone were tested in earlier experiments? With the possibility that results could be different between sexes, this could impact results

We now acknowledge a possible effect of sex in the discussion (L289-291). In our new experiment (figure S3), we tested medium contamination resulting from male or female Pe-infected individuals. Although our statistical power was too limited to properly test the effect of sex on transmission, we could not observe an overwhelmingly different pattern depending on sex. Anyways, using males for the indirect infections in our experiments was mainly for technical/practical reasons:

- 1) Because we demonstrated that the transmission of Pe to the offspring was entirely killing the next generation (see fig 3), having females for the infections means that infected vs sham vials would be different in two ways (having Pe, vs no Pe, and having no larvae vs no larvae). This confounding factor was delayed by one day, using males for infections.
- 2) Mated females lay eggs, and the crawling larvae tend to make mortality measurements hard, as the vials get dirty.
- 3) Larvae tend to eat weakened adult flies

Ideally, we would have used virgin females, but one could still argue a possible effect of reproduction (see Gupta et al 2013, cited in the ms).

- Gut load measurements: Is it known for sure that none of the bacteria has gone to other tissues or become systemic? From the methods, it seems that the bacteria were measured from whole flies, not just guts

It has been shown in previous studies (e.g. Vijendravarma, R. K., Narasimha, S., Chakrabarti, S., Babin, A., Kolly, S., Lemaitre, B. and Kawecki, T. J. (2015). Gut physiology mediates a trade-off between adaptation to malnutrition and susceptibility to food-borne pathogens. *Ecology Letters* 18, 1078–1086.) that the loss of the gut integrity was a sign of imminent death of the individuals. Therefore, it is likely that the alive individuals we sampled still had functional gut barrier and were only infected in the gut, but we cannot be absolutely certain of that. We now acknowledge this limitation in the methods (L369-372).

- Figure S3 comes before S2 in text, recommend swapping to avoid confusion

Correct. We made the requested change.

- Table S3/S4: confusing to use nipagin when it is mp in the rest of the text

We changed all instances of “nipagin” into “mp” in suppl tables.

- Line 372: should be "six-day-old individuals"

We made the requested change (L374).

- For oral infection experiments (particularly figure 2), what were the ages and mating statuses of the flies?

This information is already indicated in the methods. Flies were 6 day-old mated females at time 0 in figure 2 (L374).

Re: Spectrum03065-24R2 (Standard ingredient of *Drosophila* medium reduces transmission and virulence of the gut pathogen *Pseudomonas entomophila*)

Dear Dr. Youn Henry:

Thank you for the privilege of reviewing your work. Below you will find my comments, instructions from the Spectrum editorial office, and the reviewer comments.

Both reviewers noted that you adequately addressed their concerns, although reviewer 2 has some additional comments for you to consider. I would add that replication in time is a necessity for any published work at Spectrum, and therefore encourage you to ensure that every experiment you report on includes reported findings for at least two experiments in time. The norm for many experiments with *Drosophila* pathogenesis is often hundreds of flies, and it appears your level of replication is somewhat lower than this. Smaller N can be justifiable when the effect sizes are large, but I encourage you to consider what comparable papers use as N per experiment and number of experiments in time. If it is possible to incorporate the preliminary data you indicated in your response this may help you to avoid needing to perform additional experiments. I appreciated your detailed N as listed in the figure legends, and encourage you to repeat that, if you choose to resubmit (and I hope you will).

I hope you find these reviews encouraging, the reviewers were both very supportive of this work.

Revision Guidelines

Sincerely,
John Chaston
Editor

Reviewer #1 (Comments for the Author):

The authors did a satisfactory job addressing my and the other reviewer's feedback. I have no further comments.

Reviewer #2 (Comments for the Author):

Broadly, the authors have done a thorough job in the revisions and addressed most of the concerns from the first reviews. The issues with the transmission experiment have been addressed pretty well, with appropriate changes to language. The new experiments and images do help substantially. There are a few new and remaining points below:

- In regards to replication levels: It does help that different experiments generally agree as the authors noted, but fly infection experiments can be highly variable and typically multiple experimental blocks (ie, experiments repeated under the same conditions on different days) are done because they often do change under the same set of controlled conditions. Many stochastic aspects or less controlled variables of the experiment can change from day to day that affect the outcome (like different nutritional content from different batches of food, minor variations in time of day of infection, or minor differences in the density of the microbial culture, etc.). So, it is quite common for there to be a statistically significant effect of experimental block or day. Repeating experiments like these under the same conditions are important to ensure repeatability and reliability, as significant differences can disappear or appear depending on uncontrolled or stochastic variables. While repeating everything may be beyond the scope of this paper, it would help to address this in the discussion, even if briefly, and to include additional experimental blocks in future experiments. The authors dovetail with this issue in L271-275 and 284-287, so this may be a good place in the text to mention the caveat that one block was done for each experiment.
- Authors mention a Figure S5 in the response document, but there are only 4 supplemental figures- is this a typo?
- Figure S4- there is quite a bit of variability between the two controls, and there is a statistically significant difference between them. Could the authors hypothesize why this may be and include it in the discussion? The data appears to indicate that mp significantly helps with offspring survival on its own, even without infection

Dear Editor, below we address, point-by-point, comments from you and reviewers. Our responses are in blue and start with "###". We carefully considered all mentioned issues and corrected the manuscript accordingly.

In this submission, we provide a clean version of the manuscript, but we also show the manuscript with tracked changes since the last submission to appreciate the changes.

Editor comment:

Both reviewers noted that you adequately addressed their concerns, although reviewer 2 has some additional comments for you to consider. I would add that replication in time is a necessity for any published work at Spectrum, and therefore encourage you to ensure that every experiment you report on includes reported findings for at least two experiments in time. The norm for many experiments with *Drosophila* pathogenesis is often hundreds of flies, and it appears your level of replication is somewhat lower than this. Smaller N can be justifiable when the effect sizes are large, but I encourage you to consider what comparable papers use as N per experiment and number of experiments in time. If it is possible to incorporate the preliminary data you indicated in your response this may help you to avoid needing to perform additional experiments. I appreciated your detailed N as listed in the figure legends, and encourage you to repeat that, if you choose to resubmit (and I hope you will).

In this revised manuscript, we included two new datasets, replicating two main findings of this work (see our reply to reviewer 2 for details). Other results, although not exactly independently replicated with identical protocols, were cross validated by multiple similar experiments with consistent outcomes (e.g. figure 1 and figure S1; figure 4 and figures S4+S5). We hope that these new elements are convincing enough to strengthen our conclusions. The number of flies used, although not matching some studies (but comparable to some others), was sufficient to generate solid statistical support using the credible intervals overlapping approach (a conservative approach compared to p-values with 0.05 threshold).

Reviewer #1 (Comments for the Author):

The authors did a satisfactory job addressing my and the other reviewer's feedback. I have no further comments.

Reviewer #2 (Comments for the Author):

Broadly, the authors have done a thorough job in the revisions and addressed most of the concerns from the first reviews. The issues with the transmission experiment have been addressed pretty well, with appropriate changes to language. The new experiments and images do help substantially. There are a few new and remaining points below:

- In regards to replication levels: It does help that different experiments generally agree as the authors noted, but fly infection experiments can be highly variable and typically multiple experimental blocks (ie, experiments repeated under the same conditions on different days) are done because they often do change under the same set of controlled conditions. Many stochastic aspects or less controlled variables of the experiment can change from day to day that affect the outcome (like different nutritional content from different batches of food, minor variations in time of day of infection, or minor differences in the density of the microbial culture, etc.). So, it is quite common for there to be a statistically significant effect of experimental block or day. Repeating experiments like these under the same conditions are

important to ensure repeatability and reliability, as significant differences can disappear or appear depending on uncontrolled or stochastic variables. While repeating everything may be beyond the scope of this paper, it would help to address this in the discussion, even if briefly, and to include additional experimental blocks in future experiments. The authors dovetail with this issue in L271-275 and 284-287, so this may be a good place in the text to mention the caveat that one block was done for each experiment.

Following the editor's suggestion, we added to the manuscript the results of an earlier (pilot) experiment replicating the results shown in figures 2A and 3A. As expected, there is some variability between the two blocks in overall level of mortality (we hypothesize it comes from food "freshness", letting more or less time to natural *Drosophila* commensal to settle or not, affecting the success of *Pe*), but the main effects are highly similar, strengthening our conclusions. The new results are now presented in the new figure S2, and adjustments were made to the main text L138:

*"All these observations were confirmed in an independent replicate experiment, except that we observed no mortality at the lowest *Pe* dose ($OD_{600} = 0.0001$), most likely due to a generally lower virulence of *Pe* in this replicate (figure S2). Such variation from experiment to experiment in overall *Pe* virulence is often observed (Liehl et al., 2006)."*

and L399:

Two independent experimental replicates were performed, with minor differences explained in figure S2.

We also replicated the entire experiment of *Pe* transmission from adult to adults, the experiment with the highest chances of being subject to random variations (as already mentioned in the manuscript). We merged the results of this new experimental block with our previous results, including the experimental block in the analysis as an additional factor, and updated figure 5 accordingly. This change had limited consequences for the text, apart from minor edits on statistical results and method parts.

With that, we believe that all findings reported in the main text are properly supported with replicated experiments, except for the CFU load measurement presented in figure 2B and 2C. Therefore, we acknowledged this caveat in line 242.

"These observations should be treated with caution, as the CFU measurements were based on a single experiment and were not independently replicated."

• Authors mention a Figure S5 in the response document, but there are only 4 supplemental figures- is this a typo?

That was a typo. We wanted to mention the figure S4. However, in this new version, the previous figure S4 is now figure S5 due to the insertion of a new figure S2, shifting the numbering of other figures.

• Figure S4- there is quite a bit of variability between the two controls, and there is a statistically significant difference between them. Could the authors hypothesize why this may be and include it in the discussion? The data appears to indicate that *mp* significantly helps with offspring survival on its own, even without infection

This is indeed an interesting effect, and we can only speculate on the reasons behind this pattern. Our main hypothesis is that ambient bacteria can have detrimental effects when

their growth is not limited by the antifungals. Similarly, we noted that -mp vials always showed faster fly development, consistent with the known beneficial effects of the presence of bacteria on fly development speed. We now briefly mention this in the Results

L195 *“Of note, we observed that -mp diets also reduced somewhat the viability of sham-infected eggs (Δ pupation success % $_{(+mp\ sham - mp\ sham)} = 25$ [16; 34]), possibly resulting from growth of ambient microorganisms that are normally controlled by mp.”*

We omit this aspect in the discussion as it is not the focus of this work.

Re: Spectrum03065-24R3 (Standard ingredient of *Drosophila* medium reduces transmission and virulence of the gut pathogen *Pseudomonas entomophila*)

Dear Dr. Youn Henry:

Your manuscript has been accepted, and I am forwarding it to the ASM production staff for publication. Thank you for making those careful revisions, those brief textual changes and additional data I think will make a big difference for readers! Your paper will first be checked to make sure all elements meet the technical requirements. ASM staff will contact you if anything needs to be revised before copyediting and production can begin. Otherwise, you will be notified when your proofs are ready to be viewed.

Sincerely,
John Chaston
Editor
Microbiology Spectrum